# Assessing target genes for homing suppression gene drive

Xuejiao Xu ⓘ ✉, Jialing Fang, Jingheng Chen, Jie Yang, Xiaozhen Yang, Shibo Hou ⓘ, Weitang Sun & Jackson Champer ⓘ ✉

## Abstract

**Gene drives are engineered alleles that bias their own inheritance in offspring, enabling the spread of specific traits throughout a population. Targeting female fertility genes in a gene drive can be an efficient strategy for population suppression. In this study, we investigated nine female fertility genes in *Drosophila melanogaster* using CRISPR-based homing gene drives. Employing a multiplexed gRNA approach to prevent the formation of functional resistance alleles, we aimed to maintain high drive-conversion efficiency with low fitness costs in female drive-carriers. Drive efficiency was assessed in individual crosses and had varied performance across different target genes. Notably, drives targeting the *octopamine β2 receptor* (*oct*) and *stall* (*stl*) genes exhibited the highest drive-conversion rates and were further tested in cages. A drive targeting *stl* successfully suppressed a cage population with a high release frequency, though suppression failed in another replicate cage with a lower initial release frequency. Fitness costs in female drive carriers were observed in test cages, impacting the overall efficiency of population suppression. Further tests on the fertility of these lines using individual crosses indicated that some fitness costs were due to maternal deposition of Cas9 combined with new gRNA expression, which would only occur in progeny of drive males when testing split drives with separate Cas9 (when mimicking cages with complete drives) but not for complete drive systems. This could enable success in complete drives with higher maternal Cas9 deposition, even if cage experiments in split drives fail. Overall, our findings identify *oct* and *stl* as promising fertility targets and demonstrate both the potential and the constraints of fertility-based suppression drives, providing empirical evidence to guide the design and assessment of more efficient population control strategies.**

**Keywords** CRISPR; Homing Gene Drive; Population Suppression; Female Fertility; Fitness Costs
**Subject Categories** Evolution & Ecology; Genetics, Gene Therapy & Genetic Disease

## Introduction

Gene drive systems are a class of selfish genetic elements existing in nature, naturally biasing their own inheritance in the population. Initially, homing endonuclease genes were proposed for engineering synthetic gene drives, but target selection was constrained by the specific sequences required for endonuclease activity. The advent of CRISPR-based nucleases such as Cas9 and Cas12a, however, has vastly expanded target options through engineered guide RNA (gRNA) sequences, catalyzing the development of diverse gene drive tools (Bier, 2022; Nolan, 2021; Kefi et al, 2024; Champer et al, 2016; Wang et al, 2024). These approaches hold promise for ecologically friendly pest control strategies, with potential applications in species conservation, agriculture protection, and disease prevention.

CRISPR Cas9-based homing gene drives operate by utilizing Cas9 nuclease to cleave a target site determined by the gRNA, copying the drive element into the target allele through homology-directed repair. This process, known as drive conversion or homing, converts heterozygotes to homozygotes in the germline, ensuring the gene drive is inherited by the majority of offspring. Consequently, the gene drive frequency increases across generations, potentially leading to fixation of the drive allele. Gene drives are generally categorized as either complete drives, where Cas9 and the gRNA are linked, or split drives, where the components are at separate genetic loci. Split drives present lower biosafety risks associated with uncontrolled releases or accidental escapes, and also allow more flexible and systematic evaluation of genomic elements through different component combinations. For these reasons, split drives are often regarded as desirable for laboratory-based gene drive assessment (Champer et al, 2019; Anderson et al, 2023; Du et al, 2024), though they can potentially also be used in the field as self-limiting systems.

Depending on application goals and ecological considerations, gene drives can be designed to either modify or suppress target populations. Modification drives aim to alter populations without significantly impacting their population size, maintaining ecological niche and balance while introducing specific traits (e.g., anti-disease (Green et al, 2023; Gantz et al, 2015) or insecticide susceptibility (Kaduskar et al, 2022)) in these populations. In contrast, suppression drives are designed to quickly reduce or eliminate target populations, directly preventing agricultural loss (Yadav et al, 2023; Meccariello et al, 2024; Ma et al, 2024), disease transmission

Center for Bioinformatics, Center for Life Sciences, School of Life Sciences, Peking University, Beijing 100871, China. ✉E-mail: xuejiao.xu@pku.edu.cn; jchamper@pku.edu.cn

(Kyrou et al, 2018; Hancock et al, 2024; Deredec et al, 2011), or other negative effects caused by invasive species (Lester et al, 2020; Gierus et al, 2022).

For suppression drives, target genes should be haplosufficient but essential. They can affect both sexes (Fuchs et al, 2021) but will be more powerful when targeting female viability or reproduction. Disrupting these genes results in female mortality or sterility, reducing offspring numbers and ultimately leading to population collapse. This type of drive tends to reach an equilibrium when the drive conversion process is not perfectly efficient, thus placing a genetic load (reflecting suppressive power) on the target population, which can still eliminate the population if it is sufficiently high. A previous study (Hammond et al, 2016) has demonstrated that homing suppression drives respectively targeting three female fertility genes in malaria vector *Anopheles gambiae*. Among these, the drive targeting *nudel* met the minimum requirement to spread in cage populations initially but ultimately failed due to fitness costs in female heterozygotes from somatic Cas9 expression and functional resistant allele formation (see below). A drive in *Drosophila melanogaster* targeting *tra* suffered from even higher fitness costs in females due to somatic Cas9 expression (Carrami et al, 2018). One of the most successful population suppression gene drives to date targeted the female-specific splicing site of the sex-determination gene *doublesex* in *A. gambiae* (Kyrou et al, 2018). Despite this promising performance, similar drives in other species have not always met expectations. For instance, dominant female sterility was observed in both *Drosophila suzukii* (Yadav et al, 2023) and *D. melanogaster* (Chen et al, 2024), and male drive homozygotes in *Anopheles stephensi* exhibited sterility (Xu et al, 2024), reducing suppression efficacy. *dsx* is also an important developmental gene, required in many somatic tissues, which can be difficult for Cas9 promoters with somatic expression. These findings underscore the need for further exploration of female fertility or viability targets to broaden the options for effective population suppression.

Resistance alleles, which arise from end-joining repair rather than homology-directed repair following Cas9/gRNA cleavage, present a challenge to gene drive efficacy (Beaghton et al, 2019). They have mismatches in the target sequences and, therefore, cannot be recognized and cut by Cas9/gRNA. These alleles, if they are "functional" (preserve target gene function), can rapidly spread through target populations under strong selection pressure, outcompeting the suppression drive. For example, in a cage trial of the suppression drive targeting *nudel* in *A. gambiae*, functional resistance outcompeted the drive (Hammond et al, 2017). Even nonfunctional resistance alleles (that will still be recessive female-sterile) can be problematic for homing suppression drives because they reduce the suppressive power of the drive. This was evident in a homing drive targeting female fertility gene *yellow-G* in *D. melanogaster*, where high fitness costs in female drive heterozygotes and high embryo resistance from maternally deposited Cas9, coupled with a modest drive conversion rate, prevented reduction of the cage population due to insufficient genetic load (Yang et al, 2022).

To combat resistance, several strategies have been proposed. To address functional resistance alleles, targeting highly conserved and functionally critical sites will more likely result in nonfunctional alleles. The female-specific exon of the sex-determination gene *doublesex* is one such target, where mutations frequently disrupt

splicing signals and protein translation in females (Kyrou et al, 2018; Yadav et al, 2023; Chen et al, 2024; Xu et al, 2024). gRNA multiplexing has also emerged as a promising method to prevent functional resistance. Simultaneous targeting of multiple sites increases the likelihood of large fragment deletions, and the conversion of all target sites into functional resistance alleles is much less likely compared to single-gRNA targets (Champer et al, 2020b; Prowse et al, 2017; Khatri and Burt, 2022). A previous study in *Drosophila* expressed four gRNAs targeting well-spread sites within the target gene, though the drive conversion rate was modest and the homing events were unstable (Oberhofer et al, 2018), while closer spaced sites appear to improve efficiency (Champer et al, 2020b). More recent approaches involved transcribing multiplexed gRNAs from a single U6 promoter with closely spaced target sites, achieving higher conversion rates and eliminating functional resistance (Yang et al, 2022; Anderson et al, 2024).

The revised design can eliminate the requirement for high drive conversion if total cleavage is still high (Faber et al, 2024), but other issues can still affect genetic load and prevent population elimination. These include undesired Cas9 activity leading to fitness costs, as well as nonfunctional resistance allele formation. One solution is to enhance Cas9 activity specifically in the germline while minimizing its expression elsewhere, thus allowing wild-type alleles to persist in non-germline tissues, which is required for female fertility. Regulating maternal deposition of Cas9 can also reduce the formation of resistance alleles in the embryo, which is often a major source of these. A recent study comparing Cas9 promoters in *D. melanogaster* identified *nanos* and *CG4415* as promising candidates for homing drives, the former lacking significant somatic expression and the latter with very low maternal deposition and only modest somatic expression (Du et al, 2024). Yet, fitness costs based on cage studies are still apparent even in drive heterozygote females utilizing the *nanos* promoter, suggesting that the *yellow-G* target gene may be required in certain ovary cells where *nanos*-Cas9 is expressed (Yang et al, 2022). Because different genes are required in different tissues with potentially variable Cas9 expression, it may be possible to select alternate target genes that result in substantially reduced fitness costs.

In this study, we selected nine female fertility or viability genes and tested them using split homing gene drives to identify promising candidates with enhanced efficiency. We combined several gRNA lines, each containing four multiplexed gRNAs, with Cas9 lines under different germline promoters to evaluate drive efficiency. The most promising target genes were further assessed in cage studies. Additionally, fecundity and fertility were evaluated through single-pair crosses to investigate the causes of fitness costs. Our findings highlight potential factors limiting suppression drive efficacy and identify promising candidates for the next generation of efficient suppression gene drives.

## Results

### Target selection and expression profiling

Nine genes annotated as recessive and essential for female viability/fertility were selected from Flybase as suppression drive targets, including *intersex* (*ix*), *nudel* (*ndl*), *NADPH oxidase* (*nox*), *Octopamine β2 receptor* (*oct*), *stall* (*stl*), *transformer* (*tra*), *virilizer*

(*vir*), *defective chorion* (*dec*) and *sex-lethal* (*sxl*). Seven of these genes are located in autosomes, while *dec* and *sxl* are on the X chromosome. Among these genes, *sxl*, *tra*, and *vir* are required for sex-determination or dosage compensation, whereas the others play important roles in female-specific development and reproduction (Table EV1).

Most of these genes, with the exception of *dec* and *tra*, have homologs across various insect species, including major agricultural pests and disease vectors. The *tra* gene has been identified in *Drosophila melanogaster* and two cockroach species, *Periplaneta americana* and *Blattella germanica*, while *dec* appeared to be restricted to *D. melanogaster* and *Ceratitis capitata*. Several conserved regions were found in these proteins, suggesting conserved functions and that these regions would be potentially suitable for gRNA targeting as part of a pest control strategy across diverse insect species (Table EV2).

We analyzed the potential functions and expression profiles of these genes at different developmental stages and in adult cell types (Table EV1; Fig. EV1). Transcriptome data from Flybase show that *ndl*, *dec*, and *stl* share a similar expression profile, being highly expressed in female adults. This is consistent with their critical roles in eggshell formation, maternally affected egg patterning, and ovarian follicle development. *ix*, *sxl*, *tra* and *vir* are clustered together, with expression spread throughout developmental stages and slightly higher in early and middle-stage embryos, likely due to their significant roles in sex-determination and differentiation. Most of these genes show higher expression in female adults than male adults, except for *oct*, which exhibits slightly higher expression in males. Although *oct* is important for female ovulation and fertilization, its expression is observed from the late embryo stage, peaks at the late pupa stage, and decreases at the adult stage. This male-biased expression may result from the use of virgin females in the dataset, as *oct* is likely upregulated after mating. Moreover, *oct* is also expressed in non-reproductive tissues such as the nervous system, which may contribute to sex-specific differences in expression (Graveley et al, 2011). While *oct* may have multiple functions, it is possible that it is only essential for female fertility. In adults, *ndl*, *nox*, *tra*, *ix*, and *vir* show similar expression patterns, while *oct*, *stl*, *dec* and *sxl* are clustered together. At the cellular expression level, *ndl*, *nox*, *tra*, *ix*, and *vir* exhibit similar expression patterns, while *oct*, *stl*, *dec*, and *sxl* display a comparable pattern. In general, these genes have lower, but not absent, expression in female and male germline cells compared to other somatic cell types. Although the expression profiles suggest potential roles of these genes in males, their primary functions are likely more pronounced in females.

## Drive construct design and transformation

In this study, gRNA constructs were knocked into the female fertility genes to generate drive lines, which were then combined with existing Cas9 lines in a "split drive" design. Each gRNA construct consists of two homology arms flanking target sites for homology-directed repair, a DsRed marker driven by the eye-specific 3xP3 promoter for phenotyping, and a set of four gRNAs under the control of the U6:3 promoter (Fig. 1A). To avoid potential recombination caused by repetitive genomic promoter sequences of multiple U6-gRNA cassettes, tRNAs were placed at the start of the gRNA transcripts and between gRNAs, allowing

gRNAs to be expressed in a single transcript and spliced out to create mature gRNAs. The purpose of this gRNA multiplexing design was to prevent the formation of functional resistance alleles, and also to potentially compensate for low-efficiency gRNAs, thus increasing cut rates and drive conversion efficiency.

In these drives, target sites are cleaved, and subsequent homing or resistance allele formation occurs in germline cells in the presence of gRNA and Cas9 (after crossing gRNA lines to Cas9 lines), while targets in somatic cells ideally remain intact. When flies carrying both gRNA and Cas9 are crossed with wild-type, their offspring inherit either the gRNA "drive" allele, a disrupted resistance allele, or a wild-type allele from the drive parent. However, these heterozygotes or wild-type homozygotes could also undergo additional cleavage from the maternally deposited Cas9, converting any wild-type alleles (including those from the non-drive parent) into resistance alleles. In drive heterozygote individuals with Cas9, it may be possible for additional cleavage to occur in somatic cells, depending on Cas9 expression. Because null alleles for these target genes are recessive, females lacking wild-type alleles (carrying only drive and/or nonfunctional resistance alleles) will be sterile (Fig. 1B).

To ensure the disruption of gene function in resistance alleles, two specific criteria were adopted. First, all gRNA target sites were chosen either at the start or in the middle of essential conserved domains (Fig. 2) to reduce the chance that resistance alleles lacking frameshifts were functional (Hou et al, 2024). Additionally, multiplexing gRNAs were applied to further reduce functional resistance alleles generated in a single site. The four gRNA sites are placed closely together, with the length between two outer gRNAs no larger than 128 bp, to maximize drive conversion rates (Champer et al, 2020b). Note that two gRNA constructs (*tra-v1* and *tra-v2*) were generated to target *tra* with different gRNA sets, while only one construct was designed for each of the other genes.

We inserted the drive construct into the target site of each gene and generated transgenic lines through embryonic injection. Drive heterozygotes and homozygous flies, confirmed with genotyping, were randomly collected and respectively crossed to $w^{1118}$ files to assess their fertility. The results showed that all drive heterozygotes, as well as male homozygotes with drives targeting *nox*, *oct*, *stl*, *ndl*, and *ix* were fertile, while female homozygotes for these drives were sterile, consistent with expectations based on FlyBase annotations. Notably, we were unable to generate viable male or female homozygous adults for the drive targeting *vir*, nor were we able to generate any female homozygotes for *ix*, suggesting potential lethality during early developmental stages of these homozygotes. While female lethality is not likely to substantially affect suppression effectiveness, male homozygous lethality would reduce suppressive power similarly to X-linked targets (and haplodiploid species). Overall, we successfully generated ten multiplexed gRNA-expressing transgenic fly lines targeting nine female fertility genes in *D. melanogaster*.

## Drive efficiency assessment

To assess the drive efficiency, flies heterozygous for both drive and Cas9 alleles (receiving at least one of these alleles from a male parent) were crossed to the $w^{1118}$ line, after which the drive inheritance in their offspring were recorded (Fig. 3A; Dataset EV1). The results showed varied drive performance targeting different

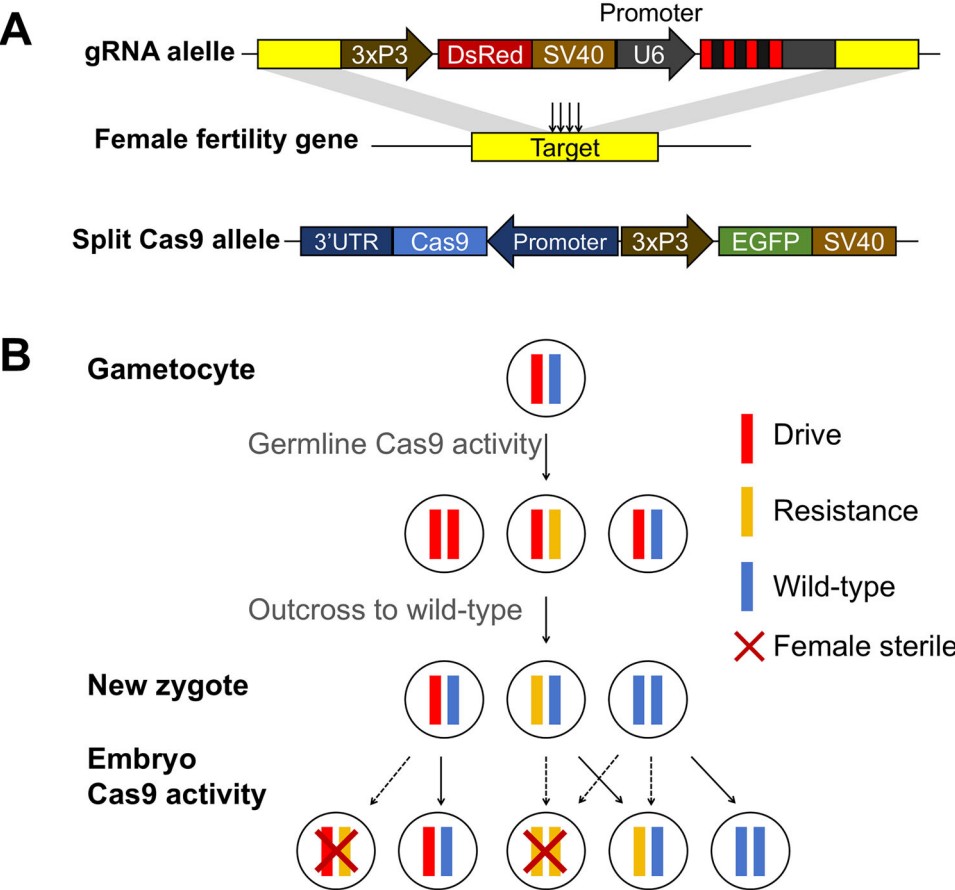

**Figure 1. Female fertility homing suppression drive construct design and mechanism.**

(A) The gRNA (drive) construct was inserted into a female fertility gene, containing the 3xP3-DsRed-SV40 marker and a set of 4-gRNA under the control of the U6 promoter to target the insertion site. A split Cas9, under the control of a germline promoter (*nanos* or *CG4415*), was provided at an unlinked site together with 3xP3-EGFP-SV40. (B) In the germline, wild-type alleles are cut by Cas9/gRNA and converted into drive alleles by homology-directed repair or resistance alleles via end-joining. Maternally deposited Cas9 and gRNA can also cut and disrupt the wild-type allele in early embryos. Females carrying any combination of drive or nonfunctional resistance alleles will be sterile, while a single copy of wild-type is sufficient to preserve female fertility. However, somatic Cas9 expression can partially or completely convert the wild-type allele to a drive or resistance allele in drive/wild-type heterozygotes.

genes. When combined with *nanos*-Cas9, the drive inheritance rates were usually higher than the 50% Mendelian expectation, except for drive females targeting *vir* and *sxl*, which were slightly lower than 50%. This could be explained by potential fitness costs from the drive insertion at these two genomic sites, or potentially an important function of these two genes in the germline, where they will be disrupted by the drive or resistance allele. The drive lines targeting *oct* and *stl* showed the highest drive inheritance rates in both females ($93.2 \pm 0.8\%$ and $94.0 \pm 1.2\%$, respectively) and males ($91.0 \pm 2.0\%$ and $89.2 \pm 1.1\%$, respectively). The drive inheritance rates of constructs targeting *ndl* and *nox* were moderate, achieving around 84-88%. When targeting *tra*, we originally tested the 4-gRNA construct *tra-v1*. However, the drive inheritance rate was relatively low (61–66%), and sequencing revealed that only the middle two gRNAs were active (Table EV3). Lack of cleavage at the outmost sites is particularly detrimental to achieving high drive conversion (Champer et al, 2020b; Chen et al, 2025; López Del Amo et al, 2020). Therefore, a second construct *tra-v2* was tested that retained the two active gRNAs and included

two new gRNAs. It showed substantially improved drive inheritance (81–83%). Notably, when targeting *ix*, the fertility of drive females was greatly reduced, and fewer offspring were produced, likely due to the fitness cost from embryo activity of Cas9/gRNA forming recessive lethal alleles. Because both *dec* and *sxl* are X-linked genes, we only assessed drive conversion in females. However, only the drive line targeting *dec* showed a biased drive inheritance in their offspring ($73.3 \pm 2.0\%$).

As a negative control, we examined the inheritance rates of drives by crossing heterozygous drive carriers with $w^{1118}$ flies in the absence of Cas9. Generally, this resulted in no bias in drive inheritance, except for the drive targeting *dec*, which was somewhat lower than the 50% Mendelian expectation ($41.3 \pm 2.6\%$, $p = 0.0012$, $z$-test) (Dataset EV2). This outcome was likely due to a fitness cost associated with the insertion of the drive construct or haploinsufficiency of *dec* in drive carriers.

Our previous study showed that the *CG4415* promoter combined with the *nanos* 3'UTR performed better with a suppression drive targeting the female fertility gene *yellow-G*,

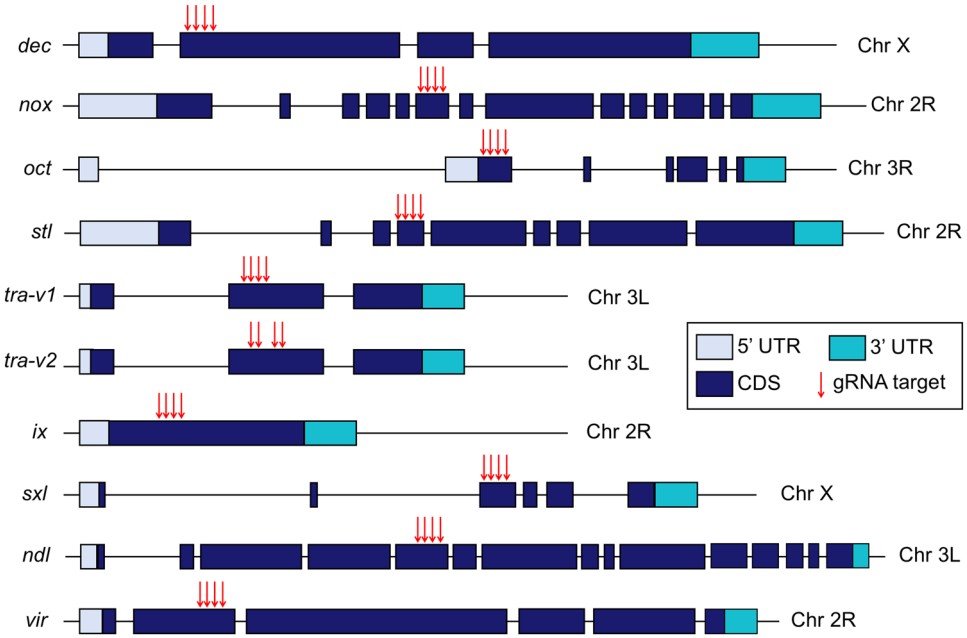

**Figure 2. Genomic structure of target genes and gRNA target sites.**

The untranslated region (UTR) and coding sequence (CDS) are shown with colored rectangles, while introns and intergenic regions are shown with black lines. gRNA target sites are indicated with red arrows. Gene elements are not drawn exactly to scale.

achieving slightly higher drive conversion and much lower embryo resistance allele formation than when Cas9 used *nanos* regulatory elements (Du et al, 2024). To assess if this was consistent across different targets, we respectively combined the *oct* and *stl* drive lines with the same two Cas9 lines (at nearly the same genomic insertion site) using the *CG4415* promoter and *nanos* 3'UTR, one of which has the orientation of the Cas9 gene reversed (Fig. 3B). When targeting *oct*, both *CG4415*-Cas9 lines showed significantly lower though still fairly high female drive conversion rates compared to the *nanos*-Cas9 line ($p < 0.0001$ and $p = 0.003$, $z$-test), although male drive conversion rates were not affected ($p = 0.8098$ and $p = 0.3797$, $z$-test). While targeting *stl*, a lower drive conversion rate was observed in females ($p < 0.0001$, $z$-test) but not males ($p = 0.8665$, $z$-test). These are consistent with the findings in the previous study that suggested somewhat lower germline expression of *CG4415*-Cas9 than *nanos*-Cas9 (Du et al, 2024), but support the potential use of these promoters with our new drive lines.

We also compared the drive performance between drive males and females in each group. The results showed higher drive conversion rates in males than females for *ix*, *tra-v1*, and *vir* with *nanos*-Cas9. This is consistent with our previous study targeting another female fertility gene *yellow-G* (Yang et al, 2022). A similar trend was also observed for drives targeting *oct* and *stl* when Cas9 was under the control of the *CG4415* promoter. However, when targeting *stl* with *nanos*-Cas9, more offspring inherited drive alleles from drive females than males. Because the drive inheritance rates in *stl* drive males and drive females were both high, the observed difference may have been caused by random factors in the test vials. In addition, no significant differences were found between drive males and drive females in other groups (Dataset EV1). These results confirm successful gene drive activity in nearly all of the

tested lines, with varying efficiencies across different target genes and Cas9 promoter combinations.

## Resistance allele formation

Embryo resistance formation rates of drives were also assessed by crossing drive females to non-drive males (Fig. 3C; Dataset EV1). Because mutation of most target genes does not show observable phenotypes, the measurement of embryo resistance of these lines was based on the percentage of sterile drive females derived from drive mothers (which should be caused by embryo resistance in nearly all cases, though this might produce a slight overestimate due to natural sterility). Note also that this defines embryo resistance as sufficient to induce sterility, but these may be mosaic rather than complete resistance. Further, the multiplex gRNA design in males may allow for continued drive conversion with a complete (as opposed to mosaic) embryo resistance allele, if some sites remain wild-type. The embryo resistance rates of drives targeting *oct*, *dec*, *stl* and *ndl* were moderate to high, ($63.3 \pm 8.8\%$, $56.7 \pm 9.1\%$, $50.0 \pm 9.1\%$, $44.4 \pm 16.56\%$, respectively), while the rate of the *nox* drive was relatively lower ($26.7 \pm 8.1\%$). Sterile drive females were randomly selected for sequencing, which showed mutations at some or all of the targets (Table EV3).

The exception was for the *tra-v2* drive, which as noted, had 100% sterility. When *tra* drive only (without Cas9) flies were intercrossed, phenotyping and genotyping showed that drive homozygous females were morphologically male and sterile, while phenotypes of heterozygous males and females were identical to $w^{1118}$ flies. Notably, in the offspring of *tra* drive female crosses, more male-like flies were observed than females (Dataset EV1). Some of these male-like flies exhibited slightly bigger body sizes like wild-

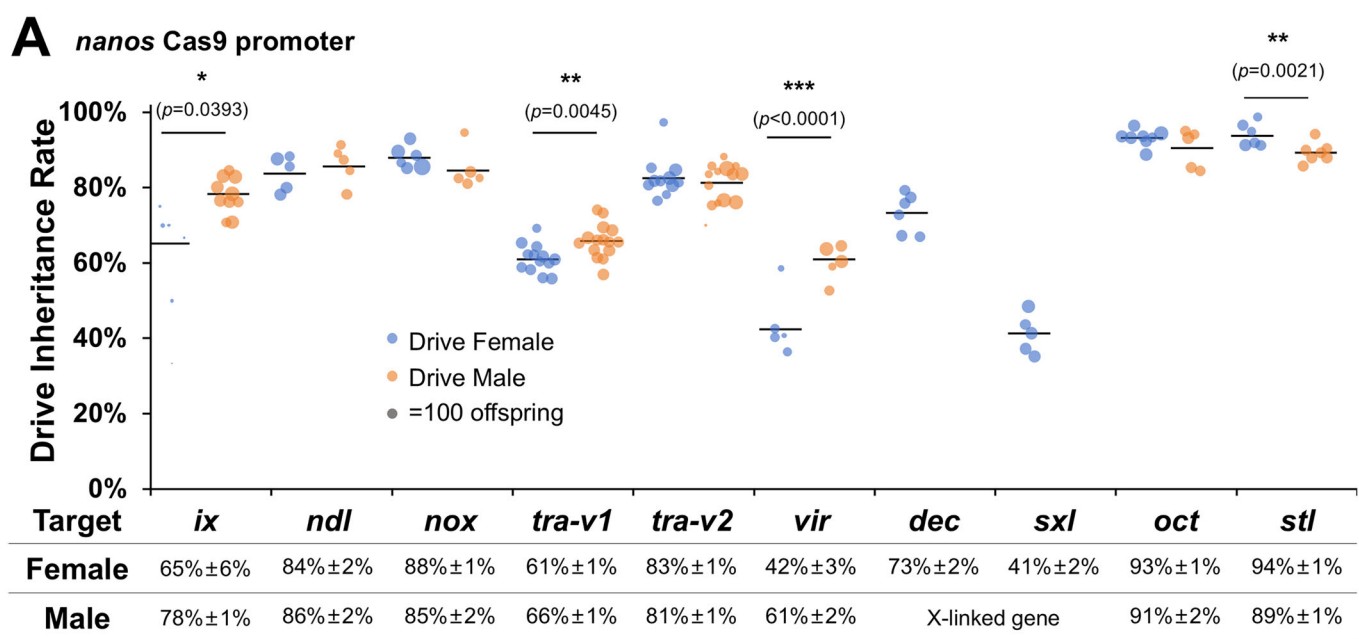

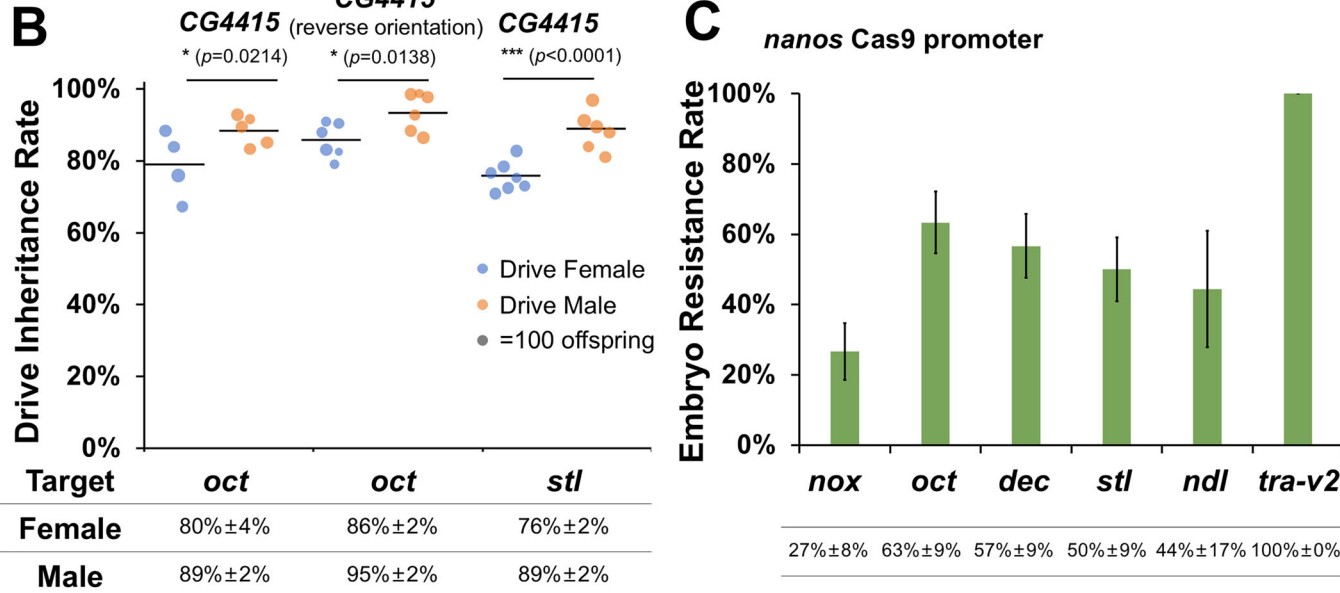

**Figure 3. Drive inheritance rates with different combinations of drive and Cas9 lines.**

(A) Drive inheritance with Cas9 driven by the *nanos* promoter. (B) Drive inheritance with Cas9 driven by the *CG4415* promoter. *CG4415* (reverse orientation) indicates that Cas9 was oriented in the same direction as the fluorescence marker, reversed from other constructs in this study. It is at nearly the same site as the *nanos*-Cas9 drive (both in chromosome 2R, but with 275 bp between the insertion sites), while the other *CG4415*-Cas9 is on chromosome 2L. Drive inheritance rate is the percentage of offspring with DsRed fluorescence from crosses between drive heterozygotes (which are also heterozygotes for Cas9) and $w^{1118}$ flies. Each dot represents the offspring from a single vial, and the size of the dots is proportional to the total number of offspring. Drive inheritance rates between males and females in each target gene are compared. For the significant differences in both (A, B), *$p < 0.05$, **$p < 0.01$, or ***$p < 0.001$ (z-test), while non-significant differences are not shown in this figure. Detailed $p$ values and sample sizes for each group are provided in Dasaset EV1. (C) Embryo resistance rate indicates the proportion of sterile drive female offspring derived from a mother that had the drive. The error bar indicates the standard error of the mean.

type females, and they were genetically female based on genotyping. These flies were recorded as "strong masculinization". Additionally, there were some flies displaying an intermediate phenotype between male and female (patchy dorsal fragment), which were recorded as "mild masculinization" (Fig. EV2). We genotyped a

random sample of males with bright eyes (indicating that they were more likely to be homozygous) with an apparent male phenotype, and their genetic sex could be determined by the presence or absence of a Y chromosome gene *PP1Y2*, which is only found in genetic males. Ten genetically female flies with the drive and strong

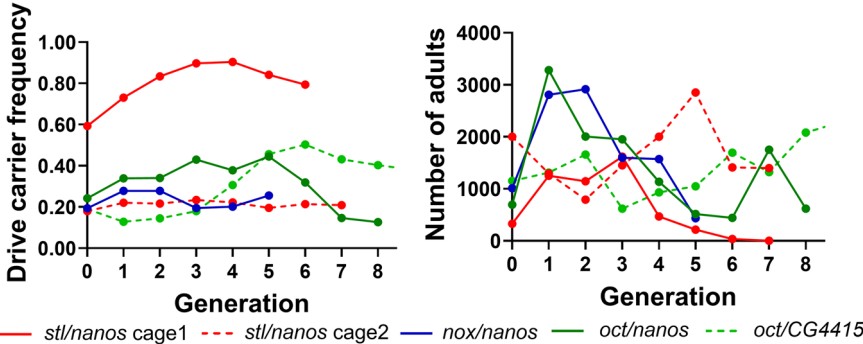

**Figure 4. Drive carrier frequency in cage populations.**

Flies carrying one copy of the drive and two copies of Cas9 were released into Cas9 homozygous populations. All flies in each discrete generation were phenotyped. In the legend, the first gene is the drive target, and the second is used for Cas9 promoter and 5′ UTR regulatory elements. See Dataset EV3 for later generations of the *oct/CG4415* cage.

masculinization and mild masculinization phenotypes were individually crossed to $w^{1118}$, and none of them produced offspring, suggesting sterility. Sequencing confirmed the presence of embryo resistance alleles in them, though wild-type was always present as well for the ones with a mild masculinization phenotype. Note that these masculinized flies represented all the genetically female offspring of drive female crosses, with the small number from male drive crosses (with a strong masculinization phenotype) likely misidentified due to the difficulty of separating strong masculinization females and wild-type males.

Additionally, the fraction of fertile drive female controls (derived from the drive father) was 90–100%. Therefore, the sterility observed in the above lines was most likely caused by embryo resistance. However, note that moderate or heavy mosaic embryo resistance could still cause sterility, so the rate of "complete" embryo resistance (mutations in all cells due to cleavage in early embryo cells) is likely lower than our reported values. This is particularly apparent in the *tra* drive, in which only the stronger masculinization phenotype had any chance of being a complete embryo resistance allele. As a performance measurement, this is therefore not entirely complete. Some females that remain fertile may still have enough mosaic cleavage to have reduced fertility. On the other hand, males that experience enough mosaic cleavage to render females sterile may be unaffected and still be able to perform normal drive conversion in the germline (Champer et al, 2019). These findings suggest moderate to high levels of embryo resistance from maternally deposited Cas9 and gRNA in the most efficient drives targeting different female fertility genes. Such resistance can reduce overall system performance, even though the resistance alleles were nonfunctional.

## Suppression drive cage experiments

We selected some of the drive lines showing superior performance in small-scale drive efficiency tests (*nox*, *oct*, and *stl*) and further assessed them in cage experiments. To set up cages, males heterozygous for the drive allele and homozygous for the Cas9 allele were generated and crossed with Cas9 homozygous females. These females were subsequently mixed with additional Cas9 females from the same batch that had instead been mated with Cas9

males. The eggs from all these Cas9 females represented the G0 generation. The drive carrier frequency and total fly number of the following generation was tracked. The drive carrier frequency was expected to increase over several generations, reaching a high equilibrium frequency and causing population suppression after forming high numbers of sterile female homozygotes.

Five cages were set up in total, including one cage combining *nanos*-Cas9 with the drive targeting *oct*, one cage containing *nanos*-Cas9 and the drive targeting *nox*, two replicate cages containing *nanos*-Cas9 and the drive targeting *stl*, and another cage containing *CG4415*-Cas9 and the drive targeting *oct*. Most of these cages were set up with a drive carrier frequency of 18–24%, except for one of the *stl* cages (cage 1), which started at 59% drive carrier frequency, while their population sizes varied from 329 to 2000 (Fig. 4; Dataset EV3).

In cage 1 for *stl*, the drive carrier frequency increased to 90% by the fourth generation and then was slightly reduced in subsequent generations. During this time, the population size expanded at first but then quickly dropped after the third generation, with complete population elimination by the seventh generation. Though the drive carrier frequency was only at a moderately high level, nonfunctional resistance alleles likely also contributed to the fraction of sterile females. However, when released at a lower frequency in cage 2, the drive frequency remained approximately constant at ~20%, with no apparent effect on the population size (Fig. EV4). The *nox* drive had a similar performance.

In the cage with *nanos*-Cas9 and drive targeting *oct*, the drive carrier frequency marginally increased but did not reach a high level. The population size of this cage fluctuated, but ultimately was not eliminated by the drive. When combined with the Cas9 element driven by *CG4415* promoter, the *oct*-targeting drive carrier frequency gradually rose up to 50%, after which it slowly declined to ~33%, with the population size apparently unaffected (Dataset EV3).

Our result indicates that with a higher initial release ratio, the cage population could be completely eliminated. However, this suppression capability was diminished when the release ratio was reduced, and the population size increased. Most likely, the high release frequency drive, even though its frequency would perhaps have declined in the long run to a lower equilibrium, was able to remain high for long enough to place a temporary high genetic load on the population, allowing its elimination. Allee effects may have

contributed to this after the population size was substantially reduced (when too few flies laid eggs on the bottles, bacterial or fungal growth would often render larvae nonviable). Despite achieving a drive conversion rate of 85% to 93% in drive heterozygotes (Fig. 3), the equilibrium frequency of drive carriers in our cage populations was lower than anticipated based on a previous modeling study (Champer et al, 2020b). This suggests a fitness cost in drive carriers in our cage populations (particularly the *CG4415* cage that likely had very low embryo resistance from maternal deposition), leading to failure to eliminate the population. Our results show successful elimination of a cage population at high drive release frequency, but other attempts with lower release frequency failed due to substantial fitness costs.

## Maximum-likelihood analysis of fitness in cage populations

To assess the drive performance of our cages, a maximum-likelihood method was applied to quantify fitness components (Yang et al, 2022; Du et al, 2024; Liu et al, 2019). This model utilized a simplified design with a single gRNA and assumed no functional resistance, consistent with the expected outcome when using multiplexed gRNAs. This is because even if functional resistance forms at one or more target sites, the remaining sites can still be cleaved and disrupted, significantly reducing the likelihood of functional resistance. Drive conversion and embryo resistance allele formation were set as per our previous measurements (embryo resistance in the *CG4415* cage was set to zero, because it was lower than 5% even with a highly active gRNA element (Du et al, 2024)). Fitness costs were assumed to occur in female drive heterozygotes due to cleavage of wild-type alleles where they were needed in certain cells (either somatic or germline cells). The parameters used for modeling can be found in Table EV4.

The estimated fitness of drive female heterozygotes in cages targeting *nox* and *oct* with the *nanos* promoter were 0.262 and 0.245, respectively. When combined with the *CG4415* promoter, the cage of drive targeting *oct* exhibited a slightly higher fitness of 0.318. For the drive targeting *stl* with the *nanos* promoter, the inferred fitness was 0.205 in the cage with a lower initial release ratio. However, in another successfully suppressed *stl* cage with a higher initial release size, fitness was improved to 0.681, with the upper limit of the 95% confidence interval reaching 1.065. The reason for this difference is unclear, but it may be related to the specific conditions of the cages. Alternatively, the improved performance and fitness in the *stl* cages might be due to a relatively smaller population size (Dataset EV3), causing reduced competition among larvae.

The failure of the four cages, either with *nanos* or the *CG4415* promoter, could perhaps be attributed to the low fertility of drive females within these cage populations. However, an important caveat is that our maximum-likelihood method would probably have inferred a low fitness even if the nature of the fitness cost was different (Liu et al, 2019). These results confirm and quantify the fitness costs present in our drives based on all of the tested cage populations.

## Fertility assessment

To confirm whether there was a fitness cost for drive female fertility, we conducted single-pair crosses to assess individual fecundity (indicated by the number of eggs being laid) and egg-to-adult viability (thus giving a complete picture of fertility when combined with fecundity). A specific cross scheme was designed to mimic the cage performance of drives (Fig. EV3, biparental Cas9 cross; Dataset EV4). In the founder generation, male drive heterozygotes that were homozygous for Cas9 were crossed with Cas9 homozygous females. Their offspring, including drive heterozygotes and non-drive flies (these served as the controls because they had the same environment and parents as the drive flies, but they likely had a single nonfunctional resistance allele resulting from a high germline cut rate in the drive parental germline with *nanos*-Cas9 (Du et al, 2024)), were collected for single-pair crosses. Note that these offspring contained two copies of Cas9. In comparison, we crossed drive flies (without Cas9) with Cas9 homozygotes in the founder generation (Fig. EV3, maternal/ paternal Cas9 cross; Datasets EV5 and EV6). In these crosses, offspring contained only one copy of Cas9, and a single resistance allele might be formed in the maternal cross but absent in the paternal cross. These drive and non-drive offspring were individually crossed to non-drive flies (Cas9 homozygotes in the biparental Cas9 cross and $w^{1118}$ flies in maternal/paternal Cas9 crosses), and their offspring were counted to assess parental fecundity and egg viability. Note that we did not perform a direct comparison of Cas9-only individuals and wild-type flies in this study, because previous studies found no significant fitness difference between Cas9-expressing lines and wild-type in the absence of a drive element (though we cannot exclude a small effect, it certainly could not come close to explaining our results) (Champer et al, 2020a; Langmüller et al, 2022).

For fecundity assessment, there are often large differences between fly batches (compare cross schemes, which were separate experiments), but differences in flies originating from the same vial were more likely due to drive effects (flies would experience identical rearing conditions in the same vial). Egg numbers laid per day were counted over the course of three days (Fig. EV4). In the biparental Cas9 cross, *nox* drive females laid more eggs per day ($26.92 \pm 2.17$) compared to the non-drive female controls ($18.20 \pm 1.57$) ($p = 0.0026$, *t*-test). No significant difference was found in the fecundity between drive carriers and non-drive controls in the drives targeting *oct* and *stl* (Dataset EV4). In comparison, no significant differences were found between the drive and non-drive flies in all groups for the maternal Cas9 cross and most groups for the paternal Cas9 cross (Datasets EV5 and EV6). It is noted that in the paternal Cas9 cross with the drive targeting *nox*, non-drive females laid more eggs per day ($26.67 \pm 2.96$) than drive females ($18.37 \pm 1.60$) ($p = 0.0150$, *t*-test), which is in contrast to the results of biparental and maternal Cas9 crosses. Besides possibly *nox*, however, we did not see any substantial fecundity effects that were likely caused by drive effects. The reasons for these differences are unclear and may be due to epigenetic effects from parents or grandparents reared in different environments. Random effects also cannot be ruled out.

To further assess fertility, egg-to-adult survival rates were determined from the same experiments (Fig. 5). In the biparental Cas9 cross, the egg-to-adult survival rates of drive females were lower than non-drive females in both *nox* and *oct* groups (both $p < 0.0001$, *z*-test), indicating that drive activity likely disrupted these genes in cells that were important for egg viability. In the *stl* group, both drive males ($p = 0.0319$, *z*-test) and drive females

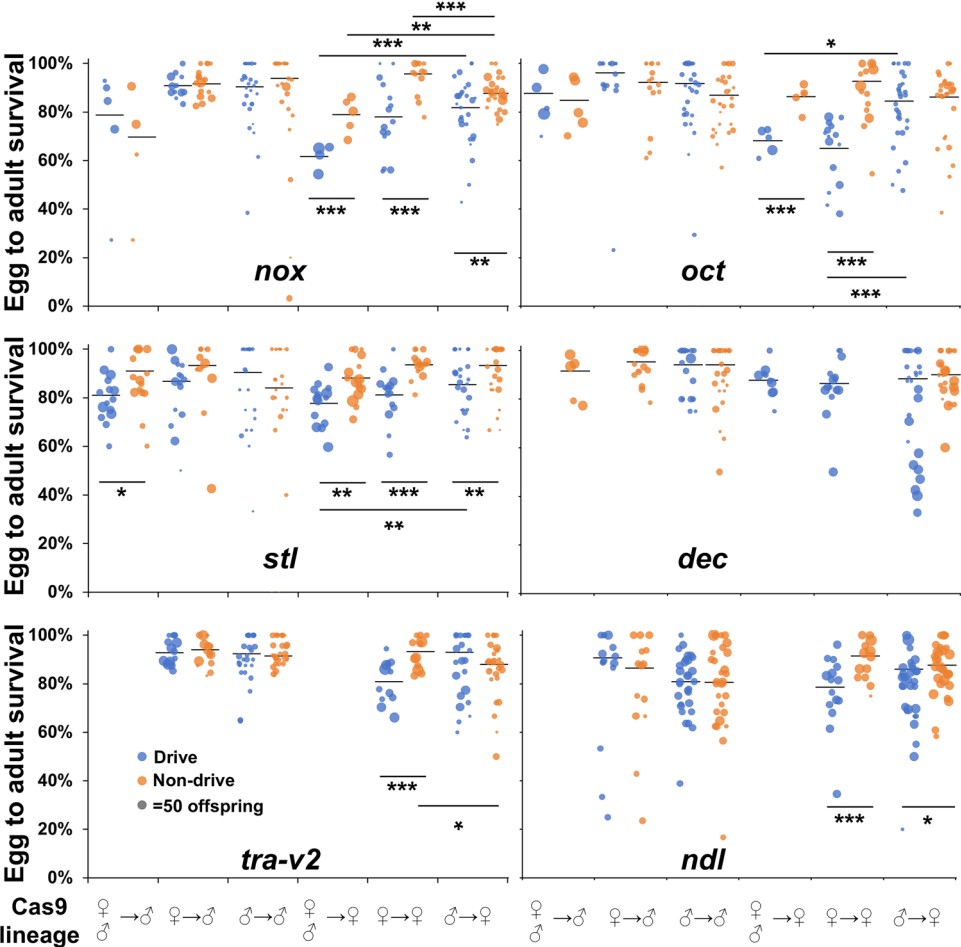

**Figure 5. Egg viability assessment of drive carriers.**

Three cross schemes were applied, as shown by the Cas9 lineage. Icons in the Cas9 lineage indicate the source of Cas9 in both grandparental (left) and parental (right) providers. For biparental Cas9, crosses were made between male drive heterozygotes and wild-type females at the drive site, but both parents were homozygous for Cas9. For male Cas9, males that were homozygous for Cas9 were crossed to drive heterozygote females (without Cas9). For female Cas9, Cas9 homozygous females were crossed to drive heterozygote males. Drive and non-drive progeny from these crosses were then crossed to non-drive flies, and the offspring were phenotyped. Biparental Cas9 cross data was not collected for *tra-v2* and *ndl*, and only partially collected for X-linked *dec*. Drive and non-drive flies are marked with dots in different colors. Significant difference is indicated with *$p < 0.05$, **$p < 0.01$, or ***$p < 0.001$ (z-test).

($p = 0.0025$, z-test) showed less egg viability compared to their non-drive controls. Considering that *stl* is essential for female fertility, which should not affect male reproduction, the reduced male egg viability was likely caused by stochastic factors existing in the Cas9 population or in the test vials (the marginally significant result would not be considered significant after multiple-testing correction). These results suggest that drive female heterozygotes in the drives targeting *nox*, *oct*, and *stl* in the cage populations suffered moderate fitness costs, reducing the genetic load of the drives and contributing to the failure to suppress the target populations. In the maternal cross, significantly lower egg viability was detected in drive females targeting *nox* ($p < 0.0001$, z-test), *oct* ($p < 0.0001$, z-test), *stl* ($p < 0.0001$, z-test) *tra-v2* ($p = 0.0002$, z-test) and *ndl* ($p = 0.0009$, z-test). Most of these results were in line with the biparental cross, while the egg viability loss in *stl* drive males in the biparental cross was not found in the maternal cross.

These fitness costs could have two potential sources. One is due to the cleavage of target genes in these drive females from newly expressed gRNA and Cas9 (which should be mostly germline-confined for *nanos*-Cas9). However, Cas9 deposition from the maternal founder in both biparental and maternal Cas9 crosses (Fig. EV5), coupled with newly expressed gRNA, may have led to Cas9 cleavage in late embryo cells, which could affect somatic cells later in development and cause significant fitness costs. Note that while this phenomenon is already incorporated in our embryo resistance studies (in which both Cas9 and gRNA were maternally deposited), it could also occur in split drives with a Cas9 mother and gRNA father, causing fitness costs even though resistance alleles do not form in the zygote and early embryo (gRNA requires time to express).

In comparison, in the paternal Cas9 cross, the egg-to-adult survival rates of drive females in *nox*, *oct*, and *stl* groups were all increased compared to the other two cross sets, though the difference between drive and non-drive females still exists in the *nox* and *oct* groups. This may be because only one Cas9 copy exists in these flies. Besides, given that these target genes are also

© The Author(s)

expressed in germ cells at some level, they may also play some roles in those cells (Fig. EV1B). Therefore, some fitness costs in our drives could still be caused by germline cleavage, which is necessary for drive activity. Although the *nanos* promoter was known to have undetectable somatic activity (Du et al, 2024; Champer et al, 2019) based on phenotyping assays and sequencing, some somatic activity may still be present, leading to fitness costs.

Note that the maternal and paternal cross setups, with only one copy of the drive and Cas9, more closely resembles a canonical complete drive design, making it more reliable in reflecting the fitness of drive females that inherit a drive from their mother or father. Therefore, we only applied these crosses to assess the fecundity and egg viability of two additional drives, *tra-v2* and *ndl*, which showed good drive performance in individual vials. The results showed that the egg viability from drive carriers did not differ from non-drive controls in the *tra-v2* group, consistent with the result of *nox* and *oct* groups. When targeting *ndl*, fewer eggs survived in drive females compared to non-drive females, while the egg viability did not vary in male crosses. A similar pattern was also observed in drives targeting *stl* and *dec*. These results indicate that Cas9 and gRNA expression from drive alleles likely affected the fitness of heterozygous drive females targeting *stl*, *dec*, and *ndl*. We also noted lower fitness in non-drive *nox* females from a biparental than paternal cross, suggesting potential haploinsufficiency (individuals from a biparental cross likely were drive/nonfunctional resistance allele heterozygotes). These results suggest that some of our measured fitness costs are associated with maternal deposition of Cas9 coupled with new gRNA expression from drive alleles.

## Discussion

Gene drives hold significant promise for population suppression, targeting applications such as disease transmission control and mitigating agricultural losses. The most effective suppression strategies typically involve disrupting sex-specific viability or fertility-essential genes, which gradually eliminate target populations over several generations in both lab and modeling tests. In this study, we evaluated nine female fertility genes using a split homing drive system in *Drosophila melanogaster*. While the drives demonstrated varying levels of efficacy, the *stl*- and *oct*-targeting drives achieved the highest drive conversion rates, with the *stl* drive successfully leading to complete population suppression in one of the cage populations. Further analyses revealed that fitness costs, driven in part by maternal Cas9 effects that would be present only in split-drive testing (e.g., crossing Cas9 females to gRNA males), significantly impacted drive performance.

The genetic load is an important parameter for understanding the suppression power of a drive. A target population will only be eliminated if the genetic load is high. Factors influencing genetic load include drive conversion efficiency, fitness costs, and nonfunctional resistance allele formation (Khatri and Burt, 2022; Zhang et al, 2024). For suppression drives, null mutants resulting from drive or resistance alleles are ultimately removed from the population due to the sterility or lethality conferred by the disrupted target gene, contributing to population sterility, but also slowing the drive by removing drive alleles in females and blocking drive conversion in males. High drive conversion rates are critical to ensuring sufficient genetic load, since this produces more

offspring carrying drive alleles in the next generation. Our drives targeting *oct* and *stl* achieved a high drive conversion rate (~80%), though still falling short of the >90% rates observed in comparable *Anopheles gambiae* systems, where suppression was achieved in a few generations despite moderate fitness costs (Kyrou et al, 2018). This discrepancy may explain why some of our cage populations were not eliminated. Optimizing drive conversion efficiency requires strong germline Cas9 activity, perhaps with appropriate timing, to enhance homology-directed repair.

Nonetheless, the successful result in the cage with high release study may point to a potential field strategy for a drive that is less efficient (perhaps even one found to be less efficient in initial field tests compared to laboratory tests). If the initial release frequency of the drive is sufficiently high and widespread, then short-term high genetic load may substantially reduce the population, perhaps enough for Allee effects to become important. At this point, even if average genetic load is slowly declining without additional drive releases, persistent moderate genetic load coupled with the Allee effect may be sufficient to ensure population elimination.

Fitness costs associated with drive carriers pose significant challenges to achieving suppression, especially when nonfunctional resistance allele formation is high (as we saw with *nanos*-Cas9 in many of our drives). We observed reduced egg viability in female drive heterozygotes when Cas9 was maternally provided (even without maternal gRNA), but this fitness loss was mitigated when Cas9 was paternal. Somatic Cas9 expression is another source of fitness cost in drive females because a lack of functional female fertility genes in somatic cells will directly impact female fertility (Hammond et al, 2021). However, the *nanos* promoter used in this study has been shown to exhibit minimal somatic activity (Yang et al, 2022; Du et al, 2024), reducing the possibility of somatic cleavage-induced fitness costs in female drive heterozygotes. Additionally, target genes used in this study are expressed and probably function in germ cells at various levels, despite our attempts to find suitable targets that were not required in the germline. Thus, loss of these essential genes in reproductive tissues could also impact reproductive egg viability and egg development. Another potential though unconfirmed source of fitness cost arises from increased cleavage events associated with multiplexed gRNAs, where the greater number of gRNAs can enhance the overall cut rate compared to single-gRNA designs (Langmüller et al, 2022).

The functions of the top-performing genes suggest a mechanistic basis for the observed fitness costs. Aside from germline cells, *nanos* has expression in other ovarian cells as well (Li et al, 2022). *CG4415* lacks this expression, but our Cas9 construct with this promoter may have a different expression pattern than the native gene, as evidenced by its support for good drive conversion in females. *stl* is essential for ovarian follicle development (Willard et al, 2004), and its disruption likely in non-germline ovary cells could compromise egg chamber development and fertility. *oct* encodes the octopamine β2 receptor, a G-protein-coupled receptor critical for ovulation and fertilization (Li et al, 2015), so if it were similarly lost, egg-laying would be directly impaired. *nox*, which encodes NADPH oxidase, contributes to calcium flux and smooth muscle contraction during ovulation (Ritsick et al, 2007), so its disruption may prevent egg-laying. *tra* is needed in the whole body for sexual development, but may also play an important role in ovarian function (Cho et al, 2018). Thus, unintended Cas9 activity at these non-germline ovary cells can directly interfere with sensitive reproductive functions,

potentially explaining the fertility costs observed in drive carriers. This issue could potentially be overcome if promoters were available that were truly restricted to germline cells rather than other reproductive cells, though it remains unclear if such promoters both exist and would retain their expression pattern at a non-native locus.

Regulating Cas9/gRNA expression is important for both drive efficiency and fitness. Because strong gRNA is usually only achieved with ubiquitously expressed Pol III promoters, choosing strong germline-specific Cas9 promoters is necessary for avoiding fitness costs from somatic expression in suppression drives. These promoters should have high germline conversion capacity to promote the inheritance of drive alleles and no embryo/somatic activity to avoid fitness costs. Our previous works showed that both *nanos* and *CG4415* have high drive conversion rates (Du et al, 2024), but *nanos* failed to suppress target populations in a homing drive targeting the female fertility gene *yellow-G* due to its fitness cost in drive females (Yang et al, 2022). *CG4415* had much lower maternal deposition, which allowed the elimination of cage populations by targeting *yellow-G* (Du et al, 2024). Here, we tested both promoters with drives targeting *oct* and *stl*, with both showing slightly higher drive efficiency than the drive targeting *yellow-G* in small-scale crosses. *CG4415* has slightly worse, though still good, performance in females, likely due to male-biased expression compared to *nanos*. However, both promoters with our new targets were unable to suppress cage populations under relatively lower release frequency. Computational analysis of cage performance indicated large fitness reductions in drive females, and the follow-up fecundity/fertility tests revealed that maternal Cas9 effects may explain some of these costs in cage experiments with *nanos*-Cas9. Because *CG4415*-Cas9 showed less maternal deposition but more somatic expression in the previous study, its fitness cost in cage populations may be a combination of somatic and germline cleavage effects, rather than a combination of maternal and germline cleavage effects like *nanos*-Cas9.

While maternal deposition poses a persistent challenge in gene drive systems, its impact is particularly pronounced in the context of split drives compared to complete drives. In our cage tests designed to simulate the spread of a complete drive, flies carrying the drive element (i.e., gRNA) were introduced into a population homozygous for the Cas9 element over several generations. This design is not fully representative of a real-world scenario where complete drives carrying both Cas9 and gRNA are released because Cas9 is expressed from a different genomic locus (Fig. EV5). Further, we showed that it also likely exaggerates fitness costs. In a split system designed to mimic a complete drive, Cas9 is always present for maternal deposition and cleavage with newly expressed gRNA, contributing to fitness costs in drive daughters. These fitness costs arise even when drive females are derived from a drive father because Cas9 will still be deposited by the non-drive mother. In comparison, in real-world releases using complete drives, Cas9 deposition will only be provided by drive mothers. Therefore, fitness costs measured with split-drive designs are likely somewhat overestimated (Fig. EV5).

In this case, switching the positions of Cas9 and gRNA (i.e., locating Cas9 in a fertility target gene and gRNA in an unlinked allele) may improve the performance of the drive in cage populations because only gRNA will be integrated into the population background. This consideration may also be important

in tethered drive (Metzloff et al, 2022; Dhole et al, 2019), with an extra complexity of applying different endonucleases (López Del Amo et al, 2022; Sanz Juste et al, 2023) at the two sites if the confined drive is also CRISPR-based. Any remaining fitness costs are likely to be more representative of a complete drive. However, because the gRNA line is specific to the target site, multiple testing of several Cas9 promoters with different split-drive targets would require a larger number of transgenic constructs. Note that for Cas9 systems with minimal maternal deposition, these considerations do not apply.

The better-performing *stl*/*nanos* cage with the higher release size showed higher fitness, which may have been improved by less competition among larvae in a lower-density population. Additionally, higher release ratios hasten the spread of the drive, quickly generating more sterile females and crashing the population, perhaps before the drive slowly declines to its natural equilibrium frequency. This indicates that for a less ideal gene drive (with high drive conversion rates but low fitness), increasing release size could help ensure a favorable outcome, at least if the release is widespread. The release size will still usually be smaller than self-limiting strategies (Zhu et al, 2024; Han and Champer, 2025).

Previous work showed that drive efficiency could be affected by different genomic positions, where Cas9 or gRNA expression pattern is possibly affected by regulatory elements or other features surrounding the insertion position (Anderson et al, 2023; Xu et al, 2022; Champer et al, 2017). Such position effects may lead to unintended somatic expression of Cas9, which can impose additional fitness costs in suppression systems. Therefore, evaluating different insertion sites may be important for optimizing drive performance. Insertion position may also influence gRNA expression, potentially affecting the performance of our drives. Additionally, our two *tra*-targeting drives, differing only in gRNA sets but sharing almost the same insertion site, displayed markedly different efficiencies, emphasizing the critical role of gRNA activity in target cleavage efficiency. Furthermore, mismatched expression patterns of Cas9 and gRNAs in split-drive systems may also contribute to reduced drive efficiency compared to full-drive systems.

For drives exhibiting low to moderate drive conversion efficiency at fertility target sites, alternative gene drive designs can be employed to enhance suppression efficacy. One promising approach involves a two-target suppression system, as we previously reported (Faber et al, 2024). In this design, drive conversion occurs at a distinct genomic site, while separate gRNAs target female fertility genes. This allows for a genetic load that is determined by the sum of the end-joining and homology-directed repair rate at the female fertility site, rather than only homology-directed repair-based drive conversion at the drive site in a standard suppression drive. By decoupling the drive element from the fertility target, this system does not require high drive conversion rates to achieve suppression, only high total mutation rates. However, this system is still vulnerable to embryo resistance and fitness costs and thus still requires suitable target sites and Cas9 promoters, even if some requirements are relaxed.

In conclusion, we successfully constructed ten gene drive systems targeting nine different female fertility genes. One drive targeting *stl* successfully eliminated a cage population using a high release frequency, possibly assisted by an Allee effect. Fitness costs observed in other cages were high and could be partially explained

by maternal deposition of Cas9 coupled with new gRNA expression, leading to reduced female fertility. This study provides insights into the use of female fertility genes for suppression of gene drive systems. Maternal Cas9 effects emerged as a key limitation, reducing genetic load and hindering suppression in cage populations. Future efforts should prioritize reducing maternal deposition while retaining low somatic expression and otherwise minimizing fitness costs. Testing of split drive could also be modified to better mimic complete systems, while also updating fitness estimates for actual deployment of self-limiting split drives. Target genes used in this study are functionally conserved and essential for female fertility, holding potential to be used in suppression gene drives in non-model insect species. By addressing these challenges, gene drives can be further refined for effective population control in a broader range of ecological and pest management contexts.

# Methods

### Reagents and tools table

| Reagent/resource | Reference or source | Identifier or catalog number |
|---|---|---|
| **Experimental models** | | |
| BHDaaN fly line | (Champer et al, 2019) | N/A |
| SNc9XnGr fly line | (Du et al, 2024) | N/A |
| SNcc9XnG fly line | (Du et al, 2024) | N/A |
| SNc9NG fly line | (Du et al, 2024) | N/A |
| *w*<sup>1118</sup> fly line | Champer Lab | N/A |
| **Recombinant DNA** | | |
| TTChsp70c9 | (Faber et al, 2024) | N/A |
| TTTgRNAt | (Yang et al, 2022) | N/A |
| TTTgRNAtRNAi | (Yang et al, 2022) | N/A |
| **Oligonucleotides and other sequence-based reagents** | | |
| PCR primers | Provided in GitHub (https://github.com/jchamper/Suppression-Targets) | BGI |
| **Chemicals, enzymes and other reagents** | | |
| NEBuilder HiFi DNA Assembly Master Mix | New England Biolabs | E2621 |
| Q5 High-Fidelity DNA Polymerase | New England Biolabs | M0491 |
| ZymoPURE II Plasmid Midiprep Kit | Zymo Research | D4201 |
| DNAzol | Thermo Fisher | 10503027 |
| **Software** | | |
| Flybase | https://flybase.org | N/A |
| NCBI database | https://blast.ncbi.nlm.nih.gov/Blast.cgi | N/A |
| MEGA12 | https://www.megasoftware.net/ | N/A |
| TBtools | (Chen et al, 2020) | N/A |
| CHOPCHOP | https://chopchop.cbu.uib.no/ | N/A |
| R Studio | https://posit.co/ | N/A |

| Reagent/resource | Reference or source | Identifier or catalog number |
|---|---|---|
| R program (3.6.1) | https://www.r-project.org/ | N/A |
| Benchling | https://www.benchling.com/ | N/A |
| **Other** | | |
| Fly transformation | Unihuaii Company | N/A |
| Microscope Fluorescence Adapter | NIGHTSEA | SFA-GR and SFA-RB-GO |

## Target gene selection and protein alignment

In this study, we focused on targeting female-specific genes for population suppression. We utilized the *Drosophila* database (https://flybase.org) to identify genes annotated as mutant female-sterile or lethal, with an expression profile biased towards somatic tissues in female adults, ideally excluding germline expression. The gene ID of selected target genes are listed in Table EV1.

To evaluate the broader applicability of our gene drive targets across species, we first assessed the evolutionary conservation of the target genes by assessing their homologs in other insect pests. Homologs of the target genes in several globally important pest species were identified by BLASTing *Drosophila* protein sequences against the NCBI database (https://blast.ncbi.nlm.nih.gov/Blast.cgi) and aligned with ClustalW. These pest species, including *Ceratitis capitata*, *Spodoptera frugiperda*, *Plutella xylostella*, *Aedes aegypti*, *Anopheles gambiae*, *Culex quinquefasciatus*, *Leptinotarsa decemlineata*, *Rhynchophorus ferrugineus*, *Periplaneta americana*, and *Blattella germanica*, belong to different insect orders and cause significant damage to human health, endangered species, and agricultural production (Table EV2; Dataset EV7).

## Expression profiling

Transcriptome data of target genes were downloaded from Flybase to analyze their expression profiles across different developmental stages (Flybase dataset: modENCODE_mRNA-Seq_development) and adult cell types (Flybase dataset: scRNAseq_2022_FCA) (Öztürk-Çolak et al, 2024). The developmental stages dataset included data from egg, larva, prepupa, pupa, adult male, and adult female, each containing various developmental time points. The adult cell types dataset was derived from single-cell transcriptomes, encompassing various cell types (e.g., neurons, somatic gonad, and germline cells). The downloaded RPKM data was first processed with a log2 transformation, defined as $\log2(RPKM + 1)$, and then clustered and plotted using TBtools software (Chen et al, 2020).

## gRNA targets design and plasmid construction

For designing gRNA target sites in *Drosophila*, we used the online tool CHOPCHOP (https://chopchop.cbu.uib.no/) to select targets with higher predicted cleavage activity. A similar multiplexed construct targeting four sites, primarily targeting the 5' coding region of essential domains, was designed for each gene. The exception was *tra*, for which two constructs (*tra-v1* and *tra-v2*)

were generated. The second construct (*tra-v2*) was developed in response to the low cleavage and conversion efficiency observed with *tra-v1*. This multiplexing approach was intended to fully disrupt the target genes and prevent the formation of functional resistance.

A similar design was employed for constructing gRNA-expressing plasmids targeting different genes. Each gRNA construct contained a DsRed marker under the control of the 3xP3 promoter, a tRNA-linked gRNA cassette driven by *D. melanogaster* U6:3 promoter, and homology arms for homology-directed repair. A step-by-step cloning method, as described in a previous study (Yang et al, 2022), was applied, and all plasmids used in this study were constructed via Gibson assembly. The reagents, including restriction enzymes, Q5 polymerases, and HiFi assembly mix, were purchased from New England Biolabs, while oligonucleotides were synthesized by BGI. Constructed plasmids were cleaned up with the ZymoPure Midiprep Kit from Zymo Research for embryo injection and confirmed by Sanger sequencing. All the final plasmid sequences are provided on GitHub (https://github.com/jchamper/Suppression-Targets).

## Fly rearing and transformation

All flies were reared according to previously described protocols (Du et al, 2024). In brief, flies were fed with artificial cornmeal medium in vials or bottles under 25 ± 1 °C with a 14/10 h day/night cycle. All work with genetically modified insects was performed following regulations approved by the biosafety office at Peking University.

To generate transgenic lines, a mixture of 500 ng/μL donor plasmid and 500 ng/μL *hsp70*-Cas9 helper plasmid was injected into $w^{1118}$ flies. The embryo injection was conducted by Unihuaii Company. After injection, the founder flies were crossed with $w^{1118}$, and their offspring were screened for red fluorescence, indicating successful transformations. Transgenic drive lines were maintained as a mixture of heterozygotes and homozygotes due to the sterility of female homozygotes.

## Fly crosses

For general drive efficiency assessment, drive carriers were crossed to Cas9 lines under the control of either *nanos* (BHDaaN) (Champer et al, 2019) or *CG4415* (SNc9XnGr or SNcc9XnG, the latter on chromosome 2L instead of 2R like the other Cas9 genes) promoters (Du et al, 2024). Their offspring, heterozygous for both drive and Cas9, were then outcrossed to $w^{1118}$. Each cross vial contained two males and four females unless specifically stated. The generated progeny were phenotyped for red and green fluorescence to assess the inheritance of drive and Cas9 alleles. For the control cross, drive heterozygous flies were crossed to $w^{1118}$ in the absence of Cas9. Their offspring were phenotyped for red fluorescence to calculate drive inheritance.

## Cage study

Due to the nature of split drives, the drive and Cas9 alleles tend to segregate after crossing. Therefore, to conduct a cage study designed to mimic the release of a complete drive in a wild population, we first integrated Cas9 alleles into the population

genomic background and then released the drive into this population (with drive individuals also containing Cas9). Specifically, drive males were outcrossed to Cas9 homozygous females for several generations to generate fly lines heterozygous for the drive and homozygous for Cas9. Males of this line were crossed with Cas9 homozygous females for two days and then removed from the bottle. The mated females, together with Cas9-only females that were mated with Cas9-only males, were equally distributed into eight food bottles for a two-day oviposition period, after which the females were discarded, and the bottles, without plugs, were placed into 25 × 25 × 25 cm cages. Flies were allowed to grow and fly inside the cages. Subsequently, food bottles were replaced with new ones after 12 days, and flies were allowed to lay eggs, which was recorded as the G0 generation. Only 1 day of oviposition was conducted in the following generations, resulting in a 13-day cycle for each generation. After oviposition, adults were frozen and collected for phenotyping, while egg-containing bottles were returned to the cages to continue the cycle of non-overlapping generations.

## Analysis of cage performance

After collecting drive carrier frequency for each generation of each cage, we employed a maximum-likelihood approach to quantify drive fitness costs. This model was similar to the model used in previous studies (Yang et al, 2022; Du et al, 2024; Metzloff et al, 2022), with a simplifying assumption of a single gRNA at the drive allele site. This assumption is necessary in the maximum-likelihood method due to the large number of genotypes in a 4-gRNA model. Here, all the target alleles would be converted into nonfunctional resistance alleles if they could not be converted into drive alleles in the germline (based on relatively high embryo resistance, we assumed that there would be few or no remaining wild-type alleles after Cas9 cleavage). The drive conversion rates and embryo resistance formation rates in males and females were set based on drive efficiency tests in individual vials. The female heterozygote fitness and effective population size were inferred by the model. Note that while this model is likely suitable for females (where a partial resistance allele renders them sterile), it may conservatively underestimate male drive conversion when such drive conversion would still be possible in individuals with partial resistance alleles. This could result in slightly underestimating fitness costs.

## Fecundity and fertility test

Three different cross schemes were conducted to investigate the fertility of drive carriers. The biparental cross mimicked the fertility performance in our cage experiment. Specifically, drive males were outcrossed to Cas9 homozygous females for several generations to produce males heterozygous for the drive and homozygous for Cas9. These males were then crossed to Cas9 homozygous females, and their offspring, including drive and non-drive individuals, were outcrossed to Cas9 homozygotes for egg number and egg viability assessment. This cross was designed to reflect the performance of drive males observed in our cage experiments.

The maternal/paternal Cas9 cross schemes were set up by crossing drive heterozygous males/females (without Cas9) to Cas9 homozygous females/males to generate flies heterozygous for both drive and Cas9 alleles or non-drive Cas9 heterozygotes. These flies were then crossed to $w^{1118}$, and the egg number and egg viability of

their offspring was recorded. These two schemes more closely reflect the performance of drives that would be deployed, though the maternal Cas9 scheme may overestimate fitness costs.

In the above cross schemes, Cas9 homozygotes or $w^{1118}$ for each target gene cross were respectively collected from the same parental bottles to minimize the batch effect caused by stochastic factors. Each cross vial contained a single pair of male and female as a biological replicate. Flies were moved to new vials every day, and their egg numbers were counted over 3 consecutive days. Progeny were reared until emerging as adults and then phenotyped. To assess the embryo resistance allele formation rate, drive females from the drive mothers were randomly collected and outcrossed to $w^{1118}$ males to assess their fertility. These females were then sequenced to confirm their genotype.

To reduce batch effects between different individuals, in each cross scheme, drive males, drive females, control males, and control females were all collected from the same parental cross (drive heterozygote crossed with Cas9 homozygote), ensuring they had the same parental effects. Additionally, because all the files were reared in the same bottle before being collected for fertility assessment, the potential effect from various population densities and food characteristics was minimized. In this assessment, control flies did not contain any drive allele. In the biparental Cas9 cross, the control flies likely had one nonfunctional resistance allele at the target site, and in the paternal/maternal Cas9 cross, the control flies had one copy of Cas9. However, this was considered necessary due to the wide variance in fertility and especially fecundity between flies from different batches. Our target genes were selected to be haplosufficient, and Cas9 alone is unlikely to have substantial fitness effects (Champer et al, 2022), so these controls likely had similar performance to wild-type flies.

### Phenotyping and genotyping

Flies anesthetized with $CO_2$ or frozen at $-20\ °C$ were phenotyped. Morphological observations were made using a stereomicroscope. Their fluorescence was confirmed using the NIGHTSEA adapter SFA-GR for DsRed and SFA-RB-GO for EGFP. The percentages of flies carrying DsRed or EGFP were recorded as the inheritance rates of drive or Cas9 alleles, respectively, in single-vial crosses.

To address potential batch effects (each vial was considered an independent batch), potentially biasing rate, and error estimates, we analyzed data as reported in previous studies (Du et al, 2024; Yang et al, 2022). We fit a generalized linear mixed-effects model with a binomial distribution (maximum-likelihood, adaptive Gauss-Hermite Quadrature, nAGQ = 25). This approach accounts for variance between batches, resulting in slightly different parameters and increased standard error estimates compared to pooling all individual progeny from different vials. The analysis was performed with the R program (3.6.1) and supported by the lme4 (1.1-21) and emmeans (1.4.2) packages.

To test the fertility of homozygotes, we collected the flies with strong fluorescence (potentially indicating homozygosity) derived from the cross of heterozygous parents, and then outcrossed them to wild-type individuals. The fertility of these flies was recorded, after which these flies were genotyped to confirm their homozygosity.

To identify resistance alleles formed in the germline or embryos, drive or non-drive flies were individually ground, and their genomic DNA was extracted using DNAzol (Thermo Fisher) according to the manufacturer's protocol. A pair of oligonucleotides were designed for each gene to amplify a 210–600 bp product covering the gRNA target region. This product was gel-cleaned, sequenced by Sanger sequencing, and analyzed with Benchling (https://www.benchling.com/). The oligonucleotide sequences are provided in GitHub (https://github.com/jchamper/Suppression-Targets).

## Data availability

This study includes no data deposited in external repositories.

The source data of this paper are collected in the following database record: biostudies:S-SCDT-10_1038-S44318-025-00683-y.

## Peer review information

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

## Acknowledgements

This project was supported by grants from the National Natural Science Foundation of China (32302455 and 32270672) and laboratory startup support from Peking University and the Center for Life Sciences. XX was supported in part by the Postdoctoral Fellowship of Peking-Tsinghua Center for Life Sciences. We thank Sam Champer for assistance with batch effect analysis.

## Author contributions

**Xuejiao Xu**: Conceptualization; Supervision; Funding acquisition; Investigation; Writing—original draft; Writing—review and editing. **Jialing Fang**: Investigation; Writing—review and editing. **Jingheng Chen**: Investigation; Writing—review and editing. **Jie Yang**: Investigation; Writing—review and editing. **Xiaozhen Yang**: Investigation; Writing—review and editing. **Shibo Hou**: Investigation; Writing—review and editing. **Weitang Sun**: Investigation; Writing—review and editing. **Jackson Champer**: Conceptualization; Supervision; Funding acquisition; Writing—original draft; Writing—review and editing.

Source data underlying figure panels in this paper may have individual authorship assigned. Where available, figure panel/source data authorship is listed in the following database record: biostudies:S-SCDT-10_1038-S44318-025-00683-y.

## Disclosure and competing interests statement

The authors declare no competing interests.

# Expanded View Figures

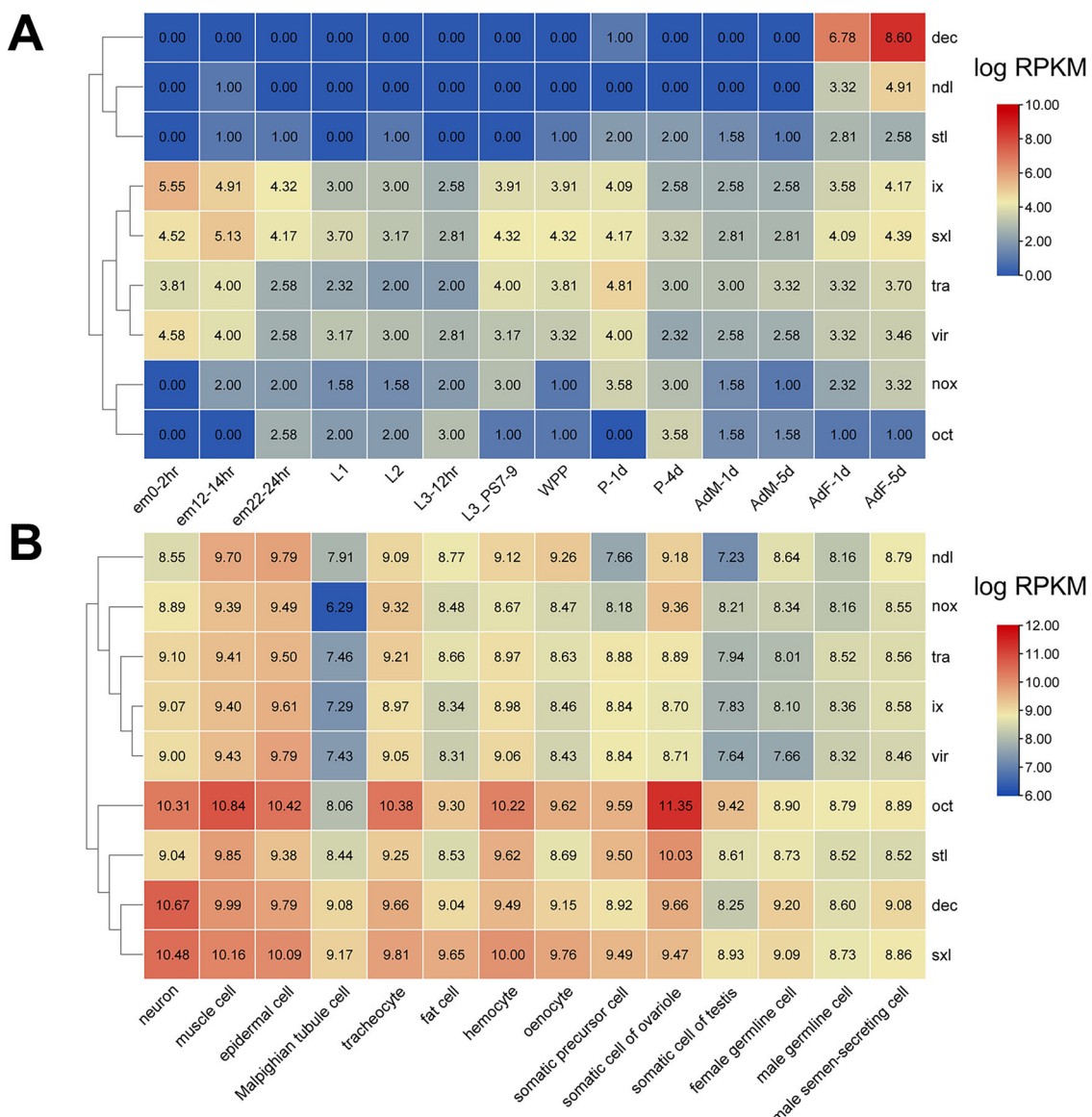

**Figure EV1. Expression profile of target genes.**

(A) Expression pattern in different developmental stages. em embryo, L larva, WPP white prepupa, P pupa, AdM adult male, AdF adult female. (B) Expression pattern in different adult cell types. The values inside the heatmap indicate expression levels of target genes (log2(RPKM + 1)). Red color indicates high expression, and blue color represents low expression. Genes with similar expression profiles are clustered together.

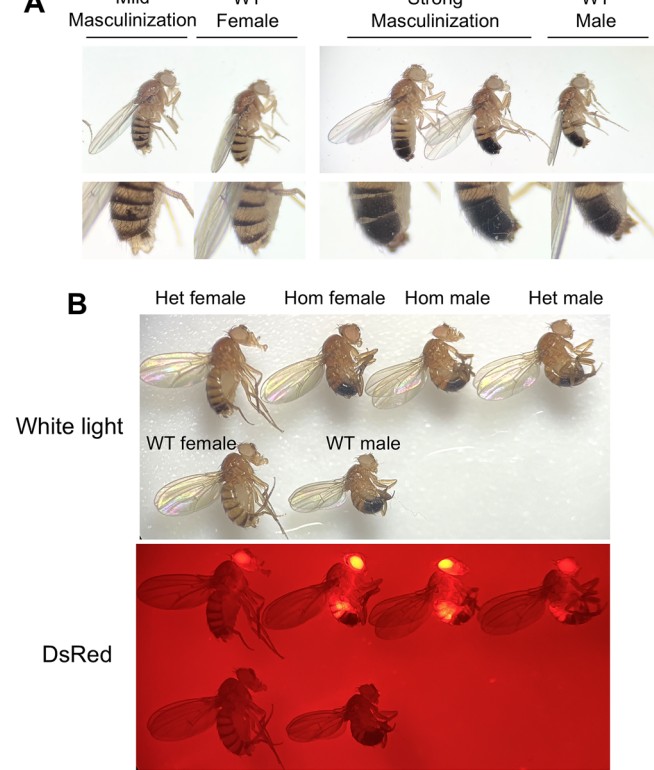

**Figure EV2. Phenotypes of drive targeting *tra*.**

(**A**) After crossing flies heterozygous for both *tra*-targeting drive and Cas9 alleles with *w^{1118}*, two abnormal phenotypes were observed in their offspring. These were genetically female but exhibited various levels of masculinization. We defined unusual patchy pigmentation in dorsal fragments as mild masculinization, while male-like flies were defined as strong masculinization. (**B**) After intercrossing drive heterozygotes (Het) in the absence of Cas9, homozygous (Hom) females showed male phenotype, while heterozygotes were identical to *w^{1118}* flies (wild-type, WT). Homozygosity is indicated by strong fluorescence and confirmed by genotyping.

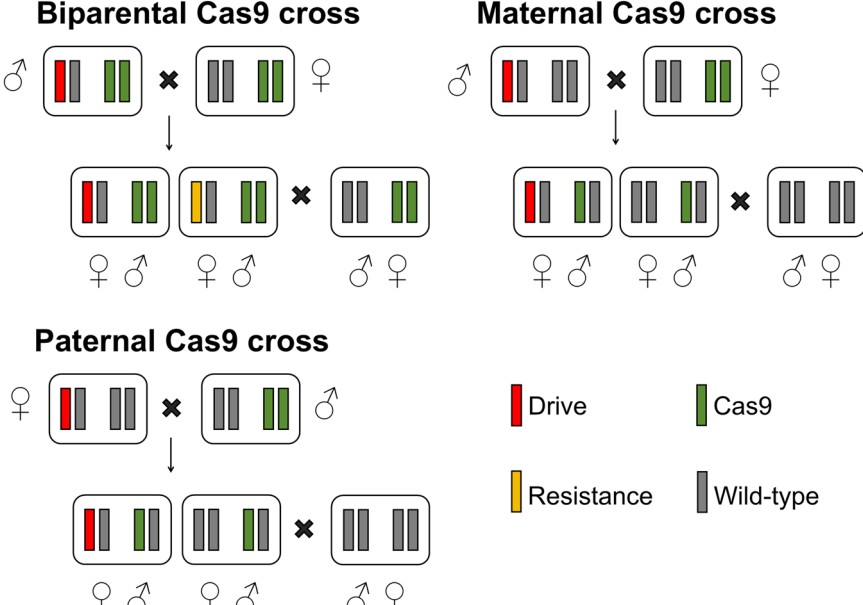

**Figure EV3. Illustration of cross schemes for fecundity and fertility tests.**

The biparental Cas9 cross was specifically designed for assessing drive fitness in cage populations. In the initial generation, males heterozygous for drive and homozygous for Cas9 were crossed to Cas9 homozygous females to generate drive and non-drive flies for fecundity and fertility assessment. In this case, the fitness of drive females was likely reduced by maternally deposited Cas9 and zygotically expressed gRNA. In maternal and paternal Cas9 crosses, Cas9 was provided by female and male flies, respectively. Biparental and maternal crosses have identical maternal effects in their female offspring, which are not present in the parental cross. Fitness costs from somatic Cas9 expression would still be present in all crosses, and these are likely higher in the biparental cross.

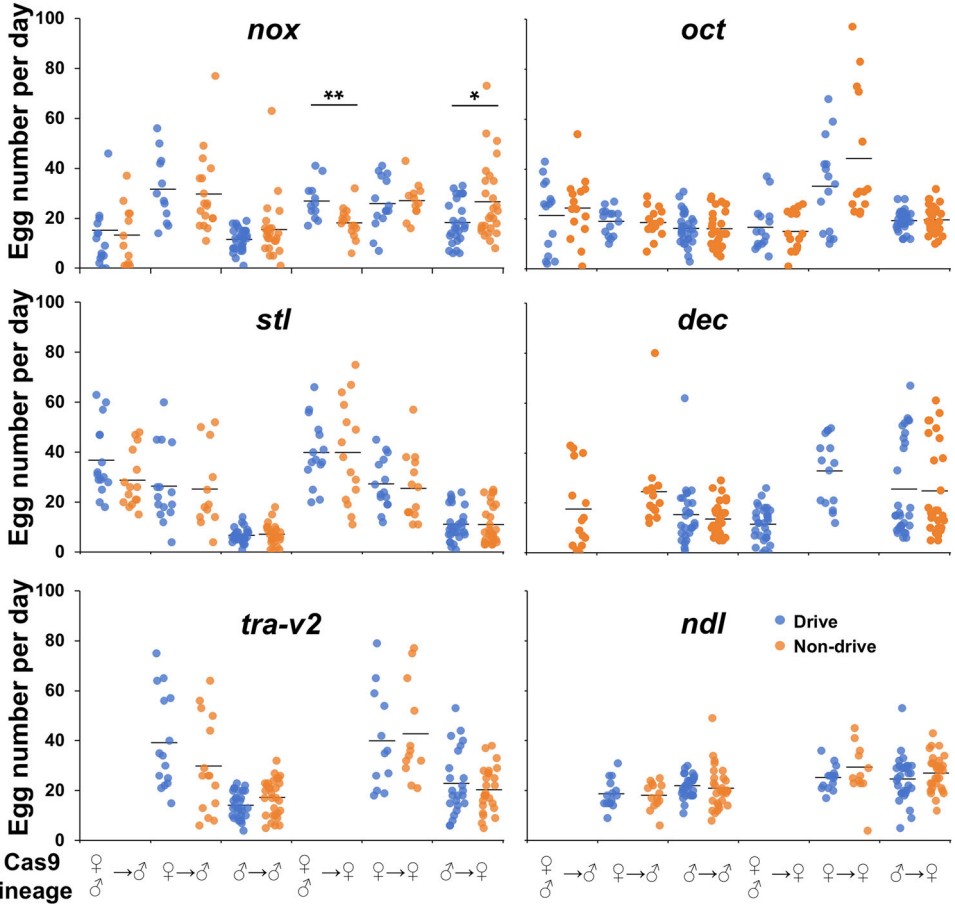

**Figure EV4. Female fecundity.**

Three cross schemes were applied, as shown by Cas9 lineage. For biparental Cas9, crosses were made between male drive heterozygotes and wild-type females at the drive site, and both parents were homozygous for Cas9. For maternal or paternal Cas9 crosses, either drive heterozygote males or females (without Cas9) were crossed to opposite sex flies that were homozygous for Cas9. For all three crosses, drive and non-drive progeny were then crossed to non-drive flies, and the offspring were phenotyped. Drive and non-drive flies are marked with dots in different colors. Biparental Cas9 cross data not collected for *tra-v2* and *ndl*, and only partially collected for X-linked *dec*. Drive and non-drive flies are marked with dots in different colors. Significant difference is indicated with *$p < 0.05$, **$p < 0.01$ (*t*-test). Raw data is provided in Datasets EV4, EV5, and EV6.

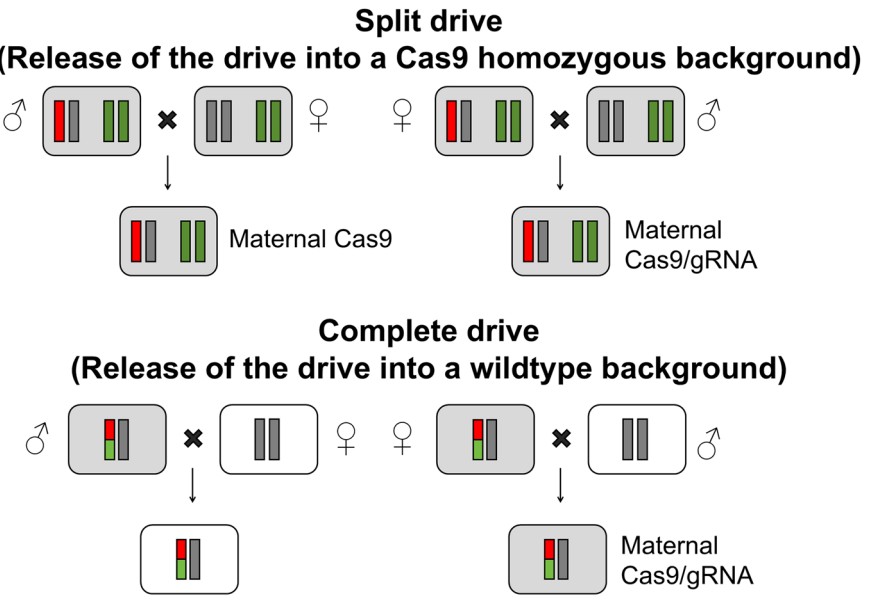

## Split drive
## (Release of the drive into a Cas9 homozygous background)

Maternal Cas9

Maternal Cas9/gRNA

## Complete drive
## (Release of the drive into a wildtype background)

Maternal Cas9/gRNA

Cas9 protein   gRNA   Cas9   gRNA+Cas9   Wild-type

**Figure EV5.  Comparison between split drive and complete mechanisms.**

In this study, flies carrying the drive element (i.e., the gRNA) were introduced into a population that is homozygous for Cas9 (upper half), representing a similar complete drive (lower half). However, there are potentially important differences. First, two copies of Cas9 are provided in all individuals for the split drive, and they are at a different genomic site unlinked to gRNA. Second, offspring will always receive maternal Cas9, which is not the case for the complete drive. This maternal Cas9 can combine with newly expressed gRNA, producing resistance allele formation or drive conversion in somatic cells. The lack of wild-type alleles in such cells can have negative fitness effects on females.

    