## [Peer Review File · The EMBO Journal]

Assessing target genes for homing suppression gene drive

Xuejiao Xu, Jialing Fang, Jingheng Chen, Jie Yang, Xiaozhen Yang, Shibo Hou, Weitang Sun, and Jackson Champer

Corresponding authors: Jackson Champer (jchamper@pku.edu.cn) , Xuejiao Xu (xuejiao.xu@pku.edu.cn)

Review Timeline:	Transferred from Review Commons:	23rd Sep 25
	Editorial Decision:	31st Oct 25
	Revision Received:	5th Nov 25
	Accepted:	26th Nov 25

Editor: Yehu Moran

Transaction Report:

This manuscript was transferred to The EMBO Journal following peer review at Review Commons.

Review #1**1. Evidence, reproducibility and clarity:****Evidence, reproducibility and clarity (Required)**

The manuscript by Xu et al. investigated split gene drive systems by targeting multiple female essential genes involved in fertility and viability in *Drosophila*. The authors evaluate the suppression efficiency through individual crosses and cage trials. Resistance allele formation and fitness costs are explored by examining the sterility and fertility of each line. Overall, the experimental design is sound and methods are feasible. The work is comprehensive, and conclusions are well supported by the data. This work offers informative insights that could guide the design of suppression gene drive systems in other invasive disease vectors or agricultural pests.

However, several points requiring clarification or improvement:

1. Methodological clarity: Some experimental details are insufficiently described, for example, regarding the setup of genetic crosses involving different Cas9 derivatives. In line 197-198, "the mated females, together with females that were mated with Cas9 only males", it is unclear whether the latter group refers to gRNA-females.
2. Regarding the inheritance rates, you included the reverse orientation of CG4415-Cas9, as I understood, it means this component is in reverse orientation with fluorescent marker. Since it is standard to design adjacent components in opposite direction to avoid transcriptional interference, the rationale for including this comparison should be better justified.
3. Embryo resistance is inferred from the percentage of sterile drive females derived from drive mothers. How many female individuals were analysed per line and why deep sequencing was not employed to directly detect resistance alleles.
4. Masculinisation phenotypes were observed upon disruption of *tra* gene. How strong intersexes were distinguished from males? What molecular markers were used to determine genetic sex. This information should be clearly provided.
5. It would be more appropriate to use "hatchability" rather than "fertility" when referring to egg-to-larva viability.
6. In cage trials, a complete gene drive is mimicked by introducing Cas9 to the background population, but this differs from actual complete gene drive, due to potential effects from separate insertion sites (different chromosome or loci). These differences could impact the system's performance and should be discussed.
7. Given the large amount of data presented, it would improve readability and interpretation

if each result section concluded with a concise summary highlighting the key findings and implications.

2. Significance:

Significance (Required)

The authors evaluate suppression efficiency through individual courses and cage trials. Resistance allele formation and fitness costs are explored by examining the sterility and fertility of each line. Overall, the experimental design is sound and methods are feasible. The work is comprehensive, and conclusions are well supported by the data. This work offers informative insights that could guide the design of suppression gene drive systems in other invasive disease vectors or agricultural pests.

3. How much time do you estimate the authors will need to complete the suggested revisions:

Estimated time to Complete Revisions (Required)

(Decision Recommendation)

Less than 1 month

Yes

Review #2

1. Evidence, reproducibility and clarity:

Evidence, reproducibility and clarity (Required)

****Paper summary****

The manuscript by Xu. et al presents an insightful and valuable contribution to the field of gene drive research. The manuscript by Xu et al. presents an insightful and valuable contribution to the field of gene drive research. The strategy of targeting and disrupting female fertility genes using selfish homing genetic elements was first proposed by Burt in

2003. However, for this approach to be effective, the phenotypic constraints associated with gene disruption have meant that the pool of suitable target genes remains relatively small - notwithstanding the significant expansion in accessible targets enabled by CRISPR-based genome editing nucleases. Population suppression gene drives are well developed as proof-of-principle systems, with some now in the late stages of development as genetic control strains. However, advancing the pipeline will require a broader set of validated target genes - both to ensure effectiveness across diverse species and to build redundancy into control strategies, reducing reliance on any single genetic target.

In their paper, the authors conduct a systematic review of nine female fertility genes in *Drosophila melanogaster* to assess their potential as targets for homing-based suppression gene drives. The authors first conduct a thorough bioinformatic review to select candidate target genes before empirically testing candidates through microinjection and subsequent *in vivo* analyses of drive efficiency, population dynamics, and fitness costs relating to fecundity and fertility. After finalising their results, the authors identify two promising candidate target genes - *oct* and *stl* - which both demonstrate high gene conversion rates and, regarding the latter, can successfully suppress a cage population at a high release frequency. However, the manuscript suffers from a lack of in-depth discussion of a key limitation in its experimental design - namely, that the authors utilise a split-drive design to assess population dynamics and fitness effects when such a drive will not reflect release scenarios in the field. The review below highlights some major strengths and weaknesses of the paper, with suggestions for improvement.

Key strengths

The study's most significant strength is in its systematic selection and empirical testing of nine distinct genes as targets for homing-based gene drive, hence providing a valuable resource that substantially expands the pool of potential targets beyond the more commonly studied target genes (e.g. *nudel*, *doublesex*, among others). The identification of suitable target genes presents a significant bottleneck in the development of gene drives and the work presented here provides a foundational dataset for future research. The authors bolster the utility of their results by assessing the conservation of candidate genes across a range of pest species, suggesting the potential for broader application.

A key finding in the paper is the successful suppression of a cage population using a *stl*-targeting gene drive (albeit at a high release frequency). This provides a critical proof-of-principal result demonstrating that *stl* is a viable target for a suppression drive. While in the paper suppression was not possible at lower release frequencies, together, the results provide evidence for complex population dynamics and threshold effects that may govern the success or failure of a gene drive release strategy - hence moving the conversation from

a technical perspective ("can it work") to how a gene drive may be implemented. Moreover, the authors also employ a multiplexed gRNA strategy for all their gene drive designs and in particular their population suppressive gene drive targeting stl. This provides further proof-of-principal evidence for multiplexed gRNAs in order to combat the evolution of functional resistance following gene drive deployment.

Finally, a further strength of this paper is in the clever dissection of fitness effects resulting from maternal Cas9 deposition. The authors design and perform a robust set of crosses to elucidate the parental source of fitness effects (i.e. maternally, paternally, or biparentally derived Cas9), finding (as they and others have before) that embryonic fitness was significantly reduced when Cas9 was inherited from a maternal source. As discussed, the authors conclude that maternal deposition is particularly pronounced in the context of split drives as opposed to complete drives, with the implication being that a complete drive might succeed where a split-drive has failed; thus providing a key directive for future study.

Concerns

The manuscript's central weakness lies in its interpretation of the results from the cage experiments - namely that a split-drive system was used to "mimic the release of a complete drive". In the study, mosquitoes carrying the drive element (i.e. the gRNA) were introduced into a population homozygous for the Cas9 element over several generations. This design is likely not representative of a real-world scenario and, as the authors state, likely exaggerates fitness costs. This is because the females carrying Cas9 will maternally deposit Cas9 protein into her eggs, with activity spanning several generations. When mated with a drive-carrying male the gRNA will immediately co-exist with maternally deposited Cas9, leading to early somatic cleavage and significant fitness costs (reflected in the author's own fertility crosses). This is fundamentally different to how a complete drive would function in a real-world release, where complete-drive males would mate with wild-type females not carrying Cas9. Their offspring would carry the drive element but would not be exposed to maternally deposited cas9, thus deleterious maternal effects would only begin to appear in the subsequent generation from females carrying the drive. Fitness costs measured from split-drive designs are therefore likely substantially overestimated compared to what would occur during the initial but critical release phase of a complete drive. This flaw weakens the paper's ability to predict the failure or success of the screened targets in a complete drive design, thus weakening the interpretation of the results from the cage trials. As a suggestion for improvement, the authors should explicitly and more prominently discuss the limitations of their split-drive model compared to complete drive models, both in the Results and Discussion. It is also recommended to include a schematic for both strategies that contrasts the experimental setup design (i.e. release of

the drive into a Cas9 homozygous background) with a complete-drive release, clearly illustrating differences in maternal deposition pathways. This will not only contextualise the results and support the author's conclusion that observed fitness costs are likely an overestimate but will further strengthen the arguments that the candidate target genes found in this study may still be viable in a complete-drive system.

A second weakness in the manuscript relates to its limited explanation and discussion of key concepts. For example, the manuscript reports a stark difference in outcome of the two *stl*-targeting drives, where a high initial release in cage 1 led to population elimination versus a failure of the drive to spread in cage 2. The authors attribute this to vague "allele effects" and stochastic factors such as larval competition; however the results appear reminiscent of the Allee effect, which is a well-characterised phenomenon describing the correlation of population size (or density) and individual fitness (or per capita population growth rate). Using their results as an example, is it plausible that the high-frequency initial release in cage 1 imposed enough genetic load to quickly drive the population density below the Allee threshold thus quickly leading to population eradication. In cage 2, the low-frequency at initial release was insufficient to cross the Allee threshold. Omitting mention of this ecological principle greatly weakens the Discussion, and further presents a missed opportunity to discuss one of the more crucial strengths of the paper - that is, in providing a deeper insight into the practical requirements for successful field implementation. In a similar vein, the authors provide only a superficial mechanistic discussion into the fitness costs associated with drives targeting key candidate genes. The paper would benefit from a deeper discussion regarding the specific molecular functions of top-performing genes (*stl*, *oct*, *nox*) and how unintended Cas9 activity could disrupt their activity, integrating known molecular functions with observed fitness costs. For instance, *oct* encodes a G-protein coupled receptor essential for ovulation and oviduct muscle relaxation, thus disruption to the *oct* gene would directly impair egg-laying which would account for the observed phenotypic effects. A deeper discussion linking unintended Cas9 activity to the specific, sensitive functions of target genes would elevate the paper from a descriptive screen to a more insightful mechanistic study.

It is curious that the authors chose two genes on the X chromosome as targets. In insects (such as *Drosophila* here) that have heterogametic sex chromosomes, homing is not possible in the heterogametic sex as there is no chromosome to home to - so there will be no homing in males. On top of that, there is usually some fitness effect in carrier (heterozygous) females, so in a population these are nearly always bad targets for drives - unless there is some other compelling reason to choose that target?

****Minor comments****

- Enhanced clarity in the Figures and data presentation would greatly improve readability. For example, Figure 5 is critical yet difficult to interpret; consider changing x-axis labels from icons to explicit text (e.g. "biparental Cas9", "maternal cas9", "paternal Cas9"). Similarly, Figure 4 is difficult to read and the y-axis label "population size" is ambiguous; consider adding shapes or dashes (rather than relying solely on colour) and clarifying the y-axis (e.g. no. adults collected) in the legend.
- Expand on or include a schematic to show the differences in construction between the tra-v1 and tra-v2 constructs to better contextualise the discrepancies in results (e.g. inheritance rates of 61%-66% for tra-v1 and 81%-83% for tra-v2 between the two).
- Minor typos e.g.:
 - Line 87: "form" to "from"
 - Line 484: "expended" to "expanded"
 - Line 560: "foor" to "for"
 - Line 732: "conversed" to "conserved"
- Clarify the split drive system: the authors introduce split drive for the first time in Line 118. They should at least give a clear definition and explanation of split drive and complete drive in the introduction.
- Line 237-238., The fitness evaluation lacks a clear description of controls. How were non-drive flies generated and validated as controls?
- Line 409-412.,line 423.,The high inheritance rates of stl and oct drives are impressive; however, variation in results across Cas9 promoters should be explained further in the discussion.
- Line 414: The CG4415 promoter yielded reduced drive conversion rates in females, yet is still referred to as a promising promoter. This conclusion seems optimistic and should be clarified/more justified.
- Specify the number of flies released, sex ratio, and cage size per generation (Line 466). This is essential for reproducibility.

2. Significance:

Significance (Required)

Overall the manuscript presents a valuable and timely resource for gene drive research, in particular for its systematic appraisal of potential target genes for population suppression drives and its rigorous assessment of the impact of maternal Cas9 deposition. The value in the generation and empirical testing of a novel multiplexed stl-targeting gene drive that led to population eradication in a cage trial should not be understated. While several key

aspects of the discussion of the manuscript should be strengthened, the study presents a meaningful contribution to the field, extending previous work and outlines important considerations for the design and implementation of effective gene drive systems.

3. How much time do you estimate the authors will need to complete the suggested revisions:

Estimated time to Complete Revisions (Required)

(Decision Recommendation)

Less than 1 month

Yes

Review #3

1. Evidence, reproducibility and clarity:

Evidence, reproducibility and clarity (Required)

In this study, Xu and colleagues explored how CRISPR-based homing gene drives could be used to suppress insect populations by targeting female fertility genes in *Drosophila melanogaster*. They engineered split gene drives with multiplexed guide RNAs to target nine candidate genes, seeking to prevent functional resistance and achieve high drive conversion with minimal fitness costs.

Here my comments about this work:

Abstract: While the stated aim of the study on line 16 is to "maintain high drive conversion efficiency with low fitness costs in female drive carriers," the conclusion in lines 29-31 shifts focus toward the broader challenges and future optimization of gene drive systems. This conclusion does not clearly highlight the specific results of the study or how they relate directly to the original objective. It would be more effective to emphasize the actual findings, such as which target genes performed best and under what conditions, and how

these findings support or contradict the stated goals. The study primarily aimed to assess the efficiency of specific female fertility genes and to evaluate strategies for minimizing the formation of functional resistance alleles, rather than proposing a protocol for optimization. Therefore, better alignment is needed between the study's aim, experimental design, and concluding statements. Clarifying this alignment would also help refine the paper's focus and more accurately communicate its contribution, including whether it is exploratory, comparative, or methodologically driven.

Introduction: One of the key design elements in this study is the use of multiplexed gRNAs. It is reasonable to assume that this strategy may influence fitness costs, potentially in more than one way. Given that assessing fitness cost is a major focus of the study, it would be helpful to include a brief discussion of previous research examining how multiplexed gRNAs may impact fitness in gene drive systems. A short review of relevant studies, if available, would provide important context for interpreting the results and could help clarify whether any observed fitness costs might be attributed, at least in part, to the multiplexing strategy itself. This addition could be appropriately placed around line 102, where gRNA design is discussed.

Line 42: Cas12a also showed efficacy using gene drives in yeast and *Drosophila*.

Line 133: The paragraph begins by stating that homologs of the target genes were identified and aligned. To improve clarity, especially for readers who are new to gene drive research, it would be helpful to begin the paragraph with a brief introductory sentence explaining the purpose of this step. For example, you could state the importance of identifying and aligning homologs to assess the conservation of target sites across species, which is critical for evaluating the broader applicability of gene drive strategies. This context would guide the reader and clarify the relevance of the analysis.

Lines 144-145: You mention that "the exception was *tra*, for which two constructs containing different gRNA sets were generated." For clarity, it would be helpful to provide a brief explanation of why two different gRNA sets were used for *tra*, and whether this differs from the approach taken with the other target genes. It's currently unclear whether all other genes were targeted using a single, standardized set of gRNAs, and this should be explicitly stated here for consistency, even though it is mentioned later in the plasmid construction section. Additionally, I suggest combining the sections on gRNA target design and plasmid construction. Since these components are closely related and sequential in the experimental workflow, presenting them together would improve the logical flow and help readers follow the methodology more smoothly.

Line 210: The analysis of the cage experiments was based on models from previous studies that used a simplified assumption of a single gRNA at the target site. While I understand this approach has precedent, it raises important questions about potential limitations. Specifically, could simplifying the analysis to one gRNA affect the conclusions of this study, given that the experimental design involves multiplexed gRNAs with four distinct target sites? The implications of using this simplified model should be clearly addressed, as the dynamics of drive efficiency, resistance formation, and fitness effects may differ when multiple gRNAs are employed. Additionally, while I am not a statistician, it is worth asking whether more sophisticated modeling approaches could be applied to account for all four gRNAs, rather than reducing the system to a single-gRNA framework. A discussion of the modeling choices and their potential consequences would strengthen the interpretation of the results.

Lines 297-300: Your results show that the expression of all target genes was higher in females, except for oct, which had higher expression in males. Additionally, oct expression decreased in adults. Given that oct is functionally important for ovulation and fertilization, processes that are primarily required in adult females, this pattern is somewhat unexpected. Could there be a possible explanation for the lower expression of oct, particularly in females and especially in adults, where its function would presumably be most critical? A brief discussion or hypothesis addressing this discrepancy would help clarify the biological relevance and interpretation of the expression data.

Lines 346-347: What is the distance between the gRNA target sites within each gene? Are all of the gRNAs confirmed to be active? It would be valuable to include a table summarizing the distance between target sites for each gene, the activity levels of the individual gRNAs, and the corresponding homing rates. This would help determine whether there is a correlation between gRNA spacing and drive efficiency. For example, Lopez del Amo et al. (Nature Communications, 2020) demonstrated that even a 20-nucleotide mismatch at each homology arm can significantly reduce drive conversion. Including such a comparative analysis in your study could provide important insights into how gRNA arrangement influences overall drive performance and would be incredibly helpful for future multiplexing designs.

Line 434: I was not able to find any sequencing data. This is important to evaluate gRNA activities and establish correlations with drive efficiency.

Line 482: Did the authors test Cas9-only individuals (without the drive) against a wild-type

population? This would help determine whether Cas9 alone has any unintended fitness effects. Additionally, is Cas9 expression stable over time and across generations? It would be helpful to include any observations or thoughts on the long-term stability and potential fitness impact of Cas9 in the absence of the drive element.

Discussion: I would appreciate a more direct and clearly stated conclusion that summarizes the key findings of the study. While the discussion addresses the main outcomes in depth, presenting a concise concluding paragraph, either at the end of the discussion or as a standalone conclusion section, would provide a stronger and more definitive closing statement. This would help reinforce what the study ultimately achieved and ensure the main takeaways are clearly communicated to the reader.

Overall, I believe this is an important study that offers valuable insights for advancing the design of CRISPR-based gene drives. The findings contribute to the development of more efficient and practical gene drive prototypes, bringing the field closer to real-world applications.

2. Significance:

Significance (Required)

In this study, Xu and colleagues explored how CRISPR-based homing gene drives could be used to suppress insect populations by targeting female fertility genes in *Drosophila melanogaster*. They engineered split gene drives with multiplexed guide RNAs to target nine candidate genes, seeking to prevent functional resistance and achieve high drive conversion with minimal fitness costs. Among the targets, the stall (*stl*) and octopamine β 2 receptor (*oct*) genes performed better, showing the highest inheritance rates in lab crosses. When tested in population cages, the *stl* drive was able to completely eliminate a fly population, but only when released at a high enough frequency, while other cages failed. These failures were traced and explained by fitness cost in drive-carrying females, caused largely by maternally deposited Cas9, which led to embryo resistance and reduced fertility. Through additional fertility assays and modeling, the team confirmed that the origin and timing of Cas9 expression, particularly from mothers, significantly impacted drive success. Surprisingly, even when Cas9 was driven by promoters with supposedly low somatic activity, such as *nanos*, fitness still persisted. The study revealed that while gene drives can be powerful, their effectiveness relies on finely balanced factors like promoter choice, drive architecture, and gene function. Overall, the research offers valuable lessons for designing robust, next-generation gene drives aimed at ecological pest control.

3. How much time do you estimate the authors will need to complete the suggested revisions:

Estimated time to Complete Revisions (Required)

(Decision Recommendation)

Between 1 and 3 months

Yes

Below, reviewer comments are in black, and our responses are in blue text.

Reviewer #1 (Evidence, reproducibility and clarity (Required)):

The manuscript by Xu et al. investigated split gene drive systems by targeting multiple female essential genes involved in fertility and viability in *Drosophila*. The authors evaluate the suppression efficiency through individual crosses and cage trials. Resistance allele formation and fitness costs are explored by examining the sterility and fertility of each line. Overall, the experimental design is sound and methods are feasible. The work is comprehensive, and conclusions are well supported by the data. This work offers informative insights that could guide the design of suppression gene drive systems in other invasive disease vectors or agricultural pests.

However, several points requiring clarification or improvement:

1 Methodological clarity: Some experimental details are insufficiently described, for example, regarding the setup of genetic crosses involving different Cas9 derivatives. In line 197-198, "the mated females, together with females that were mated with Cas9 only males", it is unclear whether the latter group refers to gRNA-females.

-We thank the reviewer for pointing out this ambiguity. The latter group refers to Cas9 females crossed to Cas9 males. We have clarified this both in the methods (line 207) and results (line 505-509).

2.Regarding the inheritance rates, you included the reverse orientation of CG4415-Cas9, as I understood, it means this component is in reverse orientation with fluorescent marker. Since it is standard to design adjacent components in opposite direction to avoid transcriptional interference, the rationale for including this comparison should be better justified.

- In our construct, 'CG4415 (reverse orientation)' indicates that Cas9 was oriented in the same direction as the fluorescent marker, while the other Cas9 constructs (nanos-Cas9 and CG4415-Cas9) places them in opposite directions. "reverse" just indicates a change from a "standard" in another study. Our previous publication showed that Cas9 orientation relative to the marker had little apparent effect on drive performance at the *yellow-G* locus. In this study, we compared both orientations in a fertility gene and again observed similar results, suggesting that orientation relative to the marker does not substantially affect drive efficiency in our system. We have clarified this in the figure legend text.

3. Embryo resistance is inferred from the percentage of sterile drive females derived from drive mothers. How many female individuals were analysed per line and why deep sequencing was not employed to directly detect resistance alleles.

-Embryo resistance can mean slightly different things for different applications. The most important is probably the fraction of females that have little to no fertility due to embryo resistance. Some of these may not have complete embryo resistance alleles, but instead, have mosaicism, with a sufficient level of resistance to still cause sterility. It is unclear exactly what proportion of resistance to wild-type may cause this, and thus, proportions from pooled

sequencing, which could include both complete and all levels of mosaicism, may not be sufficient to measure this parameter. Another relevant parameter that we did not measure is the fraction of males rendered unable to do drive conversion (this value should be closer to the complete resistance rate, but probably still lower because of the multiple gRNAs). Even in this case, deep sequencing would not allow us to determine exactly what is happening in males, making individual sequencing a preferred approach. It is very nice, of course, for characterizing which resistance alleles are present overall, but in this study, we wanted to put a bit more emphasis on the effect of resistance, rather than its sequence characterizing.

We analyzed 30 females per line for lines targeting *nox*, *oct*, *dec* and *stl*, 9 females for *ndl* and 276 individuals for line *tra-v2* (Data Set S4). We believe such individual analyses sufficiently detected embryo resistance causing sterility within reasonable error. Note that we did also randomly genotype several sterile females and found mutations at target sites that disrupted gene functions.

In response to this comment, we have added some text to justify our measurement of resistance alleles and include some of this discussion:

“Note also that this defines embryo resistance as sufficient to induce sterility, but these may be mosaic rather than complete resistance. Further, note that the multiplex gRNA design in males may allow for continued drive conversion with a complete (as opposed to mosaic) embryo resistance allele, if some sites remain wild-type.”

4. Masculinisation phenotypes were observed upon disruption of *tra* gene. How strong intersexes were distinguished from males? What molecular markers were used to determine genetic sex. This information should be clearly provided.

-We observed two types of strong masculinisation phenotypes (Figure S2), one with bigger body size than wildtype males, and the other was identical to wildtype males. The homozygosity of the drive allele could be assessed by the brightness of red fluorescence in the eyes. However, we also randomly genotyped these masculinized females (as part of a batch that included males) to confirm their sex using primers for the Y-linked gene *PP1Y2*. A specific band was detected in wild-type males but not in masculinized females, confirming their genetic sex. This information has been added to the manuscript (lines 477-480).

5. It would be more appropriate to use "hatchability" rather than "fertility" when referring to egg-to-larva viability.

-Thank you for the suggestion. We used egg-to-adult survival rates as a proxy for the fertility of their parents because they usually laid similar number of eggs. However, it still technically incorrect language. We have fixed this in line 582 and elsewhere in the section.

6. In cage trials, a complete gene drive is mimicked by introducing Cas9 to the background population, but this differs from actual complete gene drive, due to potential effects from separate insertion sites (different chromosome or loci). These difference could impact the system's performance and should be discussed.

-We appreciate this point and have added discussion on the limitations of mimicking a complete

gene drive using split components (line 766-779).

7. Given the large amount of data presented, it would improve readability and interpretation if each result section concluded with a concise summary highlighting the key findings and implications.

-Thank you for the suggestion. We have added brief summaries at the end of each results section to highlight the key findings and their significance.

Reviewer #1 (Significance (Required)):

The authors evaluate suppression efficiency through individual courses and cage trials. Resistance allele formation and fitness costs are explored by examining the sterility and fertility of each line. Overall, the experimental design is sound and methods are feasible. The work is comprehensive, and conclusions are well supported by the data. This work offers informative insights that could guide the design of suppression gene drive systems in other invasive disease vectors or agricultural pests.

-We appreciate the reviewer's positive assessment of our work.

Reviewer #2 (Evidence, reproducibility and clarity (Required)):

Paper summary

The manuscript by Xu. et al presents an insightful and valuable contribution to the field of gene drive research. The manuscript by Xu et al. presents an insightful and valuable contribution to the field of gene drive research. The strategy of targeting and disrupting female fertility genes using selfish homing genetic elements was first proposed by Burt in 2003. However, for this approach to be effective, the phenotypic constraints associated with gene disruption have meant that the pool of suitable target genes remains relatively small - notwithstanding the significant expansion in accessible targets enabled by CRISPR-based genome editing nucleases. Population suppression gene drives are well developed as proof-of-principle systems, with some now in the late stages of development as genetic control strains. However, advancing the pipeline will require a broader set of validated target genes - both to ensure effectiveness across diverse species and to build redundancy into control strategies, reducing reliance on any single genetic target.

In their paper, the authors conduct a systematic review of nine female fertility genes in *Drosophila melanogaster* to assess their potential as targets for homing-based suppression gene drives. The authors first conduct a thorough bioinformatic review to select candidate target genes before empirically testing candidates through microinjection and subsequent in vivo analyses of drive efficiency, population dynamics, and fitness costs relating to fecundity and fertility. After finalising their results, the authors identify two promising candidate target genes - oct and stl - which both demonstrate high gene conversion rates and, regarding the latter, can successfully suppress a cage population at a high release frequency. However, the manuscript suffers from a lack of in-depth discussion of a key limitation in its experimental design - namely, that the authors utilise a split-drive design to assess population dynamics and fitness effects when such a drive will not reflect release scenarios in the field. The review below highlights some major strengths and weaknesses of the paper, with suggestions for improvement.

Key strengths

The study's most significant strength is in its systematic selection and empirical testing of nine distinct genes as targets for homing-based gene drive, hence providing a valuable resource that substantially expands the pool of potential targets beyond the more commonly studied target genes (e.g. *nudel*, *doublesex*, among others). The identification of suitable target genes presents a significant bottleneck in the development of gene drives and the work presented here provides a foundational dataset for future research. The authors bolster the utility of their results by assessing the conservation of candidate genes across a range of pest species, suggesting the potential for broader application.

A key finding in the paper is the successful suppression of a cage population using a *stl*-targeting gene drive (albeit at a high release frequency). This provides a critical proof-of-principal result demonstrating that *stl* is a viable target for a suppression drive. While in the paper suppression was not possible at lower release frequencies, together, the results provide evidence for complex population dynamics and threshold effects that may govern the success or failure of a gene drive release strategy - hence moving the conversation from a technical perspective ("can it work") to how a gene drive may be implemented. Moreover, the authors also employ a multiplexed gRNA strategy for all their gene drive designs and in particular their population suppressive gene drive targeting *stl*. This provides further proof-of-principal evidence for multiplexed gRNAs in order to combat the evolution of functional resistance following gene drive deployment.

Finally, a further strength of this paper is in the clever dissection of fitness effects resulting from maternal Cas9 deposition. The authors design and perform a robust set of crosses to elucidate the parental source of fitness effects (i.e. maternally, paternally, or biparentally derived Cas9), finding (as they and others have before) that embryonic fitness was significantly reduced when Cas9 was inherited from a maternal source. As discussed, the authors conclude that maternal deposition is particularly pronounced in the context of split drives as opposed to complete drives, with the implication being that a complete drive might succeed where a split-drive has failed; thus providing a key directive for future study.

Concerns

The manuscript's central weakness lies in its interpretation of the results from the cage experiments - namely that a split-drive system was used to "mimic the release of a complete drive". In the study, mosquitoes carrying the drive element (i.e. the gRNA) were introduced into a population homozygous for the Cas9 element over several generations. This design is likely not representative of a real-world scenario and, as the authors state, likely exaggerates fitness costs. This is because the females carrying Cas9 will maternally deposit Cas9 protein into her eggs, with activity spanning several generations. When mated with a drive-carrying male the gRNA will immediately co-exist with maternally deposited Cas9, leading to early somatic cleavage and significant fitness costs (reflected in the author's own fertility crosses). This is fundamentally different to how a complete drive would function in a real-world release, where complete-drive males would mate with wild-type females not carrying Cas9. Their offspring would carry the drive element but would not be exposed to maternally deposited cas9, thus deleterious maternal effects would only begin to appear in the subsequent generation from females carrying the drive. Fitness costs measured from split-drive designs are therefore likely substantially overestimated compared to what would occur during the initial but critical release phase of a complete drive.

This flaw weakens the paper's ability to predict the failure or success of the screened targets in a complete drive design, thus weakening the interpretation of the results from the cage trials. As a suggestion for improvement, the authors should explicitly and more prominently discuss the limitations of their split-drive model compared to complete drive models, both in the Results and Discussion. It is also recommended to include a schematic for both strategies that contrasts the experimental setup design (i.e. release of the drive into a Cas9 homozygous background) with a complete-drive release, clearly illustrating differences in maternal deposition pathways. This will not only contextualise the results and support the author's conclusion that observed fitness costs are likely an overestimate but will further strengthen the arguments that the candidate target genes found in this study may still be viable in a complete-drive system.

-We sincerely appreciate the thoughtful review and the valuable comments and suggestions provided, which have helped improve both the clarity and readability of this study. We have revised several parts in the discussion of the manuscript and hope that these changes adequately address the concerns raised. We have also made Figure S5 to illustrate the differences between two release strategies (biparental-Cas9 split drive in our study and complete drive in real release).

Please note that this type of fitness cost may have partially undermined our cage study (the fitness effect is notable, but still small compared to total fitness costs), but this is also among the first studies to propose and investigate this phenomenon in the first place (it is also noted in another preprint from our lab but to our knowledge not proposed elsewhere). Thus, part of the impact of our manuscript is showing that this is important, which may inform future cage studies in our lab and elsewhere.

A second weakness in the manuscript relates to its limited explanation and discussion of key concepts. For example, the manuscript reports a stark difference in outcome of the two stl-targeting drives, where a high initial release in cage 1 led to population elimination versus a failure of the drive to spread in cage 2. The authors attribute this to vague "allele effects" and stochastic factors such as larval competition; however the results appear reminiscent of the Allee effect, which is a well-characterised phenomenon describing the correlation of population size (or density) and individual fitness (or per capita population growth rate). Using their results as an example, is it plausible that the high-frequency initial release in cage 1 imposed enough genetic load to quickly drive the population density below the Allee threshold thus quickly leading to population eradication. In cage 2, the low-frequency at initial release was insufficient to cross the Allee threshold. Omitting mention of this ecological principal greatly weakens the Discussion, and further presents a missed opportunity to discuss one of the more crucial strengths of the paper - that is, in providing a deeper insight into the practical requirements for successful field implementation.

-While we do indeed mention this Allee effect (the "allele effect" noted above is a misspelling that we have corrected), we were hesitant to give it much discussion, considering that the specific Allee effect in our cages is likely of a very different nature than one would find in nature (we explain that it is likely due to bacterial growth that occurs when fewer larvae are present). However, it is perhaps still a good excuse to cover it in the discussion, while still noting that the specific Allee effect in our cage may not be representative. We have added the following text:
 "Nonetheless, the successful result in the cage with high release study may point to a potential

field strategy for a drive that is less efficient (perhaps even one found to be less efficient in initial field tests compared to laboratory tests). If the initial release frequency of the drive is sufficiently high and widespread, then short-term high genetic load may substantially reduce the population, perhaps enough for Allee effects to become important. At this point, even if average genetic load is slowly declining without additional drive releases, persistent moderate genetic load coupled with the Allee effect may be sufficient to ensure population elimination.”

In a similar vein, the authors provide only a superficial mechanistic discussion into the fitness costs associated with drives targeting key candidate genes. The paper would benefit from a deeper discussion regarding the specific molecular functions of top-performing genes (*stl*, *oct*, *nox*) and how unintended Cas9 activity could disrupt their activity, integrating known molecular functions with observed fitness costs. For instance, *oct* encodes a G-protein coupled receptor essential for ovulation and oviduct muscle relaxation, thus disruption to the *oct* gene would directly impair egg-laying which would account for the observed phenotypic effects. A deeper discussion linking unintended Cas9 activity to the specific, sensitive functions of target genes would elevate the paper from a descriptive screen to a more insightful mechanistic study.

-We appreciate the reviewer’s comment. We have added a discussion to further explain fitness cost caused by unintended Cas9 activity disrupting target gene functions. However, keep in mind that the exact timing of Cas9 cleavage and the exact timing of these gene’s essential functions is still somewhat uncertain, which may limit insights from this line of analysis compared to a situation where ideal, high quality data is available for both of these. Here is the new material in the discussion:

“The functions of the top-performing genes suggests a mechanistic basis for the observed fitness costs. Aside from germline cells, *nanos* has expression in other ovary cells as well. CG4415 lacks this expression, but our Cas9 construct with this promoter may have a different expression pattern than the native gene, as evidenced by its support for good drive conversion in females. *stl* is essential for ovarian follicle development, and its disruption likely in non-germline ovary cells could compromise egg chamber development and fertility. *oct* encodes the octopamine β 2 receptor, a G-protein coupled receptor critical for ovulation and fertilization, so if it were similarly lost, egg-laying would be directly impaired. *nox*, which encodes NADPH oxidase, contributes to calcium flux and smooth muscle contraction during ovulation, so its disruption may prevent egg laying. *tra* is needed in the whole body for sexual development, but may also play an important role in ovary function. Thus, unintended Cas9 activity at these non-germline ovary cells can directly interfere with sensitive reproductive functions, potentially explaining the fertility costs observed in drive carriers. This issue could potentially be overcome if promoters were available that were truly restricted to germline cells rather than other reproductive cells, though it remains unclear if such promoters both exist and would retain their expression pattern at a non-native locus.”

It is curious that the authors chose two genes on the X chromosome as targets. In insects (such as *Drosophila* here) that have heterogametic sex chromosomes, homing is not possible in the heterogametic sex as there is no chromosome to home to - so there will be no homing in males. On top of that, there is usually some fitness effect in carrier (heterozygous) females, so in a

population these are nearly always bad targets for drives - unless there is some other compelling reason to choose that target?

-Our rationale for testing X-linked targets is twofold. First, these genes are likely to play important roles in sex-specific functions and may have a different expression pattern (which is why specifically *Dec* was included), potentially reducing fitness costs. Although homing cannot occur in males, if drive conversion at these sites in females is very high and fitness costs are minimal, the resulting genetic load could still be sufficient to suppress populations (thus, such candidates could be superior even in diploids if they happen to have a lower fitness costs). Second, X-linked targets may have broader relevance for suppression drives in haplodiploid pests (e.g., fire ants), which has the same population dynamics as an X-linked target in a diploid populations. Our results therefore could have provided useful insights for such scenarios (such as for fire ants: Liu et al., *bioRxiv* 2025) if drive performance was sufficient for followup testing.

Minor comments

- Enhanced clarity in the Figures and data presentation would greatly improve readability. For example, Figure 5 is critical yet difficult to interpret; consider changing x-axis labels from icons to explicit text (e.g. "biparental Cas9", "maternal cas9", "paternal Cas9"). Similarly, Figure 4 is difficult to read and the y-axis label "population size" is ambiguous; consider adding shapes or dashes (rather than relying solely on colour) and clarifying the y-axis (e.g. no. adults collected) in the legend.

-We appreciate the reviewer's comment and have revised Figure 4 as suggested. Regarding Figure 5, we attempted to replace the icons with text labels; however, this was not possible because there is very little horizontal space and two generations to specify. Instead, we have revised the figure legend to provide a clearer explanation, which can hopefully improve clarity..

- Expand on or include a schematic to show the differences in construction between the tra-v1 and tra-v2 constructs to better contextualise the discrepancies in results (e.g. inheritance rates of 61%-66% for tra-v1 and 81%-83% for tra-v2 between the two).

-We have expanded Figure 2 to compare the constructs of tra-v1 and tra-v2. The further explanation of these two constructs was added into the result section: 'When targeting tra, we originally tested the 4-gRNA construct tra-v1. However, the drive inheritance rate was relatively low (61%-66%), and sequencing revealed that only the middle two gRNAs were active (Table S3). Lack of cleavage at the outmost sites is particularly detrimental to achieving high drive conversion. Therefore, a second construct tra-v2 was tested that retained the two active gRNAs and included two new gRNAs. It showed substantially improved drive inheritance (81%-83%).'

- Minor typos e.g.:

- o Line 87: "form" to "from"

- o Line 484: "expended" to "expanded"

- o Line 560: "foor" to "for"

- o Line 732: "conversed" to "conserved"

-We have revised these typos.

- Clarify the split drive system: the authors introduce split drive for the first time in Line 118.

They should at least give a clear definition and explanation of split drive and complete drive in the introduction.

-We have included an introduction of split drive and complete drive in the introduction (line 47-53).

- Line 237-238., The fitness evaluation lacks a clear description of controls. How were non-drive flies generated and validated as controls?

-Drive heterozygotes were crossed with Cas9 homozygotes to generate the flies used for fitness evaluation. From the same cross, non-drive progeny were obtained and used as controls, ensuring they shared a comparable genetic background and rearing conditions with the drive-carrying individuals. We have now clarified in the manuscript results that “these served as the controls because they had the same environment and parents as the drive flies”.

- Line 409-412.,line 423.,The high inheritance rates of stl and oct drives are impressive; however, variation in results across Cas9 promoters should be explained further in the discussion.

-In the discussion section (lines 751-765), we included a dedicated paragraph addressing the variation observed between the nanos and CG4415 promoters. We have now expanded it to briefly note some differences:

“Our previous works showed that both nanos and CG4415 have high drive conversion rates⁸, but nanos failed to suppress target populations in a homing drive targeting the female fertility gene yellow-G due to its fitness cost in drive females²⁷. CG4415 had much lower maternal deposition, which allowed the elimination of cage populations by targeting yellow-G⁸. Here, we tested both promoters with drives targeting oct and stl, with both showing slightly higher drive efficiency than the drive targeting yellow-G in small-scale crosses. CG4415 has slightly worse though still good performance in females, likely due to male-biased expression compared to nanos.”

- Line 414: The CG4415 promoter yielded reduced drive conversion rates in females, yet is still referred to as a promising promoter. This conclusion seems optimistic and should be clarified/more justified.

-Based on our previous study cited in this context, CG4415 shows relatively lower germline conversion rates compared to nanos, although still remaining at a high level. Importantly, CG4415 also exhibits reduced maternal deposition relative to nanos, which could help mitigate fitness costs associated with maternal deposition—an important consideration for suppression systems. Taken together, while its conversion efficiency is lower (but only slightly), the potential benefits of reduced maternal deposition and perhaps even fitness costs provide a rationale for regarding CG4415 as a promising promoter. We state this when first introducing the promoter in the “Drive efficiency assessment” results subsection.

- Specify the number of flies released, sex ratio, and cage size per generation (Line 466). This is essential for reproducibility.

-We appreciate the reviewer’s comment and have revised the text to clarify our release approach, which differed from that used in other studies (which tend to have substantial fitness differences between lines in the first generation that can complicate analysis and change results). Rather than directly releasing drive males or females into cages, we first crossed drive males with

non-drive females and then mixed them with non-drive females mated to non-drive males. The offspring (including males and females) from these crosses were recorded as the G0 generation, and their ratios were recorded as release frequency. We have specified the release ratio adult numbers in the following paragraph and supplementary file.

Reviewer #2 (Significance (Required)):

Overall the manuscript presents a valuable and timely resource for gene drive research, in particular for its systematic appraisal of potential target genes for population suppression drives and its rigorous assessment of the impact of maternal Cas9 deposition. The value in the generation and empirical testing of a novel multiplexed *stl*-targeting gene drive that led to population eradication in a cage trial should not be understated. While several key aspects of the discussion of the manuscript should be strengthened, the study presents a meaningful contribution to the field, extending previous work and outlines important considerations for the design and implementation of effective gene drive systems.

-We thank the reviewer for their encouraging and constructive comments. We are pleased that the systematic evaluation of target genes, the analysis of maternal Cas9 deposition, and the multiplexed *stl*-targeting drive were recognized as valuable contributions. We have strengthened the discussion as suggested, and we believe these revisions further enhance the manuscript as an aid for the design and implementation of future gene drive systems.

Reviewer #3 (Evidence, reproducibility and clarity (Required)):

In this study, Xu and colleagues explored how CRISPR-based homing gene drives could be used to suppress insect populations by targeting female fertility genes in *Drosophila melanogaster*. They engineered split gene drives with multiplexed guide RNAs to target nine candidate genes, seeking to prevent functional resistance and achieve high drive conversion with minimal fitness costs.

Here my comments about this work:

Abstract: While the stated aim of the study on line 16 is to "maintain high drive conversion efficiency with low fitness costs in female drive carriers," the conclusion in lines 29-31 shifts focus toward the broader challenges and future optimization of gene drive systems. This conclusion does not clearly highlight the specific results of the study or how they relate directly to the original objective. It would be more effective to emphasize the actual findings, such as which target genes performed best and under what conditions, and how these findings support or contradict the stated goals. The study primarily aimed to assess the efficiency of specific female fertility genes and to evaluate strategies for minimizing the formation of functional resistance alleles, rather than proposing a protocol for optimization. Therefore, better alignment is needed between the study's aim, experimental design, and concluding statements. Clarifying this alignment would also help refine the paper's focus and more accurately communicate its contribution, including whether it is exploratory, comparative, or methodologically driven.

-We have revised the abstract to clarify the alignment as suggested by the reviewer. We note that this discrepancy is due to the initial aim of our study being different than some of the important

lessons learned along the way regarding fitness effects from Cas9 deposition in split drives. Still, we agree that it would be better to be more consistent in our wording and conclusions.

Introduction: One of the key design elements in this study is the use of multiplexed gRNAs. It is reasonable to assume that this strategy may influence fitness costs, potentially in more than one way. Given that assessing fitness cost is a major focus of the study, it would be helpful to include a brief discussion of previous research examining how multiplexed gRNAs may impact fitness in gene drive systems. A short review of relevant studies, if available, would provide important context for interpreting the results and could help clarify whether any observed fitness costs might be attributed, at least in part, to the multiplexing strategy itself. This addition could be appropriately placed around line 102, where gRNA design is discussed.

-We have added an explanation in the Discussion to mention this. However, it has not been conclusively shown that multiplexed gRNAs have any effect on fitness. Indeed, there have been some multiplexed constructs that seem to have no fitness effect, and some that have high fitness costs. This doesn't rule out the potential for multiplexed gRNAs to influence fitness itself, but it means that the mechanism may be complex. The new text reads:

"Another potential though unconfirmed source of fitness cost arises from increased cleavage events associated with multiplexed gRNAs, where the greater number of gRNAs can enhance the overall cut rate compared to single-gRNA designs."

Line 42: Cas12a also showed efficacy using gene drives in yeast and Drosophila.

-We now mention Cas12a at the beginning of the introduction.

Line 133: The paragraph begins by stating that homologs of the target genes were identified and aligned. To improve clarity, especially for readers who are new to gene drive research, it would be helpful to begin the paragraph with a brief introductory sentence explaining the purpose of this step. For example, you could state the importance of identifying and aligning homologs to assess the conservation of target sites across species, which is critical for evaluating the broader applicability of gene drive strategies. This context would guide the reader and clarify the relevance of the analysis.

-We have added the explanation as suggested.

Lines 144-145: You mention that "the exception was *tra*, for which two constructs containing different gRNA sets were generated." For clarity, it would be helpful to provide a brief explanation of why two different gRNA sets were used for *tra*, and whether this differs from the approach taken with the other target genes. It's currently unclear whether all other genes were targeted using a single, standardized set of gRNAs, and this should be explicitly stated here for consistency, even though it is mentioned later in the plasmid construction section. Additionally, I suggest combining the sections on gRNA target design and plasmid construction. Since these components are closely related and sequential in the experimental workflow, presenting them together would improve the logical flow and help readers follow the methodology more smoothly.

-We have combined both the gRNA target design and plasmid construction sections. We also discuss the two *tra* constructs early in the results section (see response to reviewer 2).

Line 210: The analysis of the cage experiments was based on models from previous studies that used a simplified assumption of a single gRNA at the target site. While I understand this approach has precedent, it raises important questions about potential limitations. Specifically, could simplifying the analysis to one gRNA affect the conclusions of this study, given that the experimental design involves multiplexed gRNAs with four distinct target sites? The implications of using this simplified model should be clearly addressed, as the dynamics of drive efficiency, resistance formation, and fitness effects may differ when multiple gRNAs are employed. Additionally, while I am not a statistician, it is worth asking whether more sophisticated modeling approaches could be applied to account for all four gRNAs, rather than reducing the system to a single-gRNA framework. A discussion of the modeling choices and their potential consequences would strengthen the interpretation of the results.

-We have clarified this. While we have modeled multiple gRNAs with high fidelity in SLiM, the maximum likelihood method is not very amenable to such treatment. It may cause our fitness estimate to be a small overestimate, but given the low fitness inferences, would certainly not have a large enough effect to fundamentally change any conclusion (and should be of a consistent level across all cages). We now discuss this in the methods section.

Lines 297-300: Your results show that the expression of all target genes was higher in females, except for oct, which had higher expression in males. Additionally, oct expression decreased in adults. Given that oct is functionally important for ovulation and fertilization, processes that are primarily required in adult females, this pattern is somewhat unexpected. Could there be a possible explanation for the lower expression of oct, particularly in females and especially in adults, where its function would presumably be most critical? A brief discussion or hypothesis addressing this discrepancy would help clarify the biological relevance and interpretation of the expression data.

-Based on transcriptome data from FlyBase, derived from Graveley et al. (2011), Oct is indeed expressed slightly higher in adult males than in adult females. This difference may be attributed to the fact that the female flies used in the study were virgins; Oct expression could be upregulated post-mating to mediate ovulation. Additionally, Oct is expressed not only in reproductive tissues but also in other organs such as the nervous system, where sex-specific differences in cell type composition or neural activity may contribute to the observed expression bias. However, high expression does not necessarily correlate with essential expression. Though Oct could have multiple functions, it's still possible that the only apparent phenotype upon knockout is female sterility. We have added the following text:

“This male-biased expression may result from the use of virgin females in the dataset, as oct is likely upregulated after mating. Moreover, oct is also expressed in non-reproductive tissues such as the nervous system, which may contribute to sex-specific differences in expression³⁸. While oct may have multiple functions, it is possible that it is only essential for female fertility.”

Lines 346-347: What is the distance between the gRNA target sites within each gene? Are all of the gRNAs confirmed to be active? It would be valuable to include a table summarizing the distance between target sites for each gene, the activity levels of the individual gRNAs, and the corresponding homing rates. This would help determine whether there is a correlation between

gRNA spacing and drive efficiency. For example, Lopez del Amo et al. (Nature Communications, 2020) demonstrated that even a 20-nucleotide mismatch at each homology arm can significantly reduce drive conversion. Including such a comparative analysis in your study could provide important insights into how gRNA arrangement influences overall drive performance and would be incredibly helpful for future multiplexing designs.

-We have showed previously that close spacing of gRNAs should help maintain high drive conversion efficiency, and this is alluded to indirectly in the introduction (we now mention it more directly). In our study, gRNAs were positioned in close proximity without overlap, with the general distance between the outermost cut sites within each gene being <128 bp. This is noted in the results section when discussing construct design.

We have added a summary table (Table S3) presenting the sequencing results, which also showed gRNA activity levels. Notably, most but not all gRNAs were active, at least for embryo resistance (low to moderate activity may still be present in the germline). Coupled with varying activity levels for those that were active, this likely contributed to reduced drive conversion due to mismatches at the homology arms. This observation supports the notion that drive performance could be optimized by selecting and arranging more active gRNAs. Consistent with this, our second construct targeting *tra* (*tra-v2*) exhibited a higher inheritance rate than the original construct, suggesting that gRNA arrangement and activity critically influence drive efficiency. Testing the activity of every single gRNA requires the construction of multiple gRNA lines, since *in vitro* or *ex vivo* tests will not be accurate as *in vivo* transformation test. However, in our study, as long as drive conversion rates were reasonably high, further optimization was not needed. Therefore, the multiplexing gRNA design can not only maximize drive conversion, but also reduce labor filtering an increased number of 1-gRNA designs with lower performance.

Line 434: I was not able to find any sequencing data. This is important to evaluate gRNA activities and establish correlations with drive efficiency.

-We have added a summary of the sequencing results in Table S3, though these are for embryo resistance alleles. Note that while high gRNA activity is correlated with high drive inheritance, these are not directly related. For suppression drives, germline resistance rates are usually of low importance compared to drive inheritance, so we did not assess these in detail (and pessimistically assumed complete germline resistance in our cage models).

Line 482: Did the authors test Cas9-only individuals (without the drive) against a wild-type population? This would help determine whether Cas9 alone has any unintended fitness effects. Additionally, is Cas9 expression stable over time and across generations? It would be helpful to include any observations or thoughts on the long-term stability and potential fitness impact of Cas9 in the absence of the drive element.

-We did not perform a direct comparison of Cas9-only individuals and wild-type flies in this study. However, previous studies (Champer et al., Nature Communications, 2020 - Langmuller et al., eLife 2022), which we now cite in the discussion, found no significant fitness difference between very similar Cas9-expressing lines and wild type in the absence of a drive element, indicating no significant fitness impact from Cas9 alone (though we cannot exclude a small effect, it certainly could not come close to explaining our results). In our experiments, Cas9 expression was

generally stable across generations, as indicated by consistent drive inheritance and fertility test results obtained from independent batches. Separate from this study, we did observe rare instability in one *nanos*-Cas9 line, which had remained stable for over five years but recently became inactive (low population maintenance size may have caused stochastic removal of the functional allele). It is something to watch out for, but probably not on the timescale of a single study.

Discussion: I would appreciate a more direct and clearly stated conclusion that summarizes the key findings of the study. While the discussion addresses the main outcomes in depth, presenting a concise concluding paragraph, either at the end of the discussion or as a standalone conclusion section, would provide a stronger and more definitive closing statement. This would help reinforce what the study ultimately achieved and ensure the main takeaways are clearly communicated to the reader.

-We have revised and expanded the last paragraph of the discussion section to make our findings more direct and clear.

Overall, I believe this is an important study that offers valuable insights for advancing the design of CRISPR-based gene drives. The findings contribute to the development of more efficient and practical gene drive prototypes, bringing the field closer to real-world applications.

Reviewer #3 (Significance (Required)):

In this study, Xu and colleagues explored how CRISPR-based homing gene drives could be used to suppress insect populations by targeting female fertility genes in *Drosophila melanogaster*. They engineered split gene drives with multiplexed guide RNAs to target nine candidate genes, seeking to prevent functional resistance and achieve high drive conversion with minimal fitness costs. Among the targets, the stall (*stl*) and octopamine β 2 receptor (*oct*) genes performed better, showing the highest inheritance rates in lab crosses. When tested in population cages, the *stl* drive was able to completely eliminate a fly population, but only when released at a high enough frequency, while other cages failed. These failures were traced and explained by fitness cost in drive-carrying females, caused largely by maternally deposited Cas9, which led to embryo resistance and reduced fertility. Through additional fertility assays and modeling, the team confirmed that the origin and timing of Cas9 expression, particularly from mothers, significantly impacted drive success. Surprisingly, even when Cas9 was driven by promoters with supposedly low somatic activity, such as *nanos*, fitness still persisted. The study revealed that while gene drives can be powerful, their effectiveness relies on finely balanced factors like promoter choice, drive architecture, and gene function. Overall, the research offers valuable lessons for designing robust, next-generation gene drives aimed at ecological pest control.

-We sincerely appreciate the reviewer's positive and thoughtful comments. We agree that the points raised highlight the importance of our findings and hope that our revisions have further improved both the clarity and overall content of the manuscript.

Dear Dr. Champer,

Thank you for transferring your manuscript with Review Commons referee reports and responses to The EMBO Journal.

We secured reports from two of the three original referees. Given the referees' positive recommendations, I would like to invite you to submit a revised version of the manuscript, addressing the remaining technical issues raised by our editorial assistance team. This is essential for allowing the acceptance of your manuscript.

We generally allow three months as standard revision time. Yet, in light of the very minor nature of the comments, I am confident you would be able to submit your revised manuscript much earlier.

Thank you for the opportunity to consider your work for publication. I look forward to your revision.

Yours sincerely,

Yehu Moran
Academic Editor
The EMBO Journal

Revision to The EMBO Journal should be submitted online within 90 days, unless an extension has been requested and approved by the editor; please click on the link below to submit the revision online before 29th Jan 2026:

Link Not Available

Specific comments by editorial assistance team

- *AUTHOR CHECKLIST: Please provide a completed author checklist, which you can download from the author guidelines.
- *FIGURES IN SEPARATE FILES: Please remove the figures from the manuscript text and upload them as separate, high resolution figure files. The legends should stay in the manuscript text and should be placed after the References section. Please rename the suppl. figures "Figure EV1" - EV 5 and also upload them as separate figure files. Place the legends in the manuscript text after the main figure legends and under the heading "Expanded View Figure Legends"
- *KEYWORDS: missing. Please provide 5 KWs.
- *DATA AVAILABILITY SECTION: To list the primary data generated in your study, we would kindly ask you to include a formal "Data availability section" (after Methods) that follows the example below:
"The datasets produced in this study are available in the following databases:
- [data type]: [full name of the resource] [accession number/identifier] ([doi or URL or identifiers.org/DATABASE:ACCESSION])"
If this data is not yet public, please make it now publicly available.
If your study doesn't include such data please use the following instead:
"This study includes no data deposited in external repositories"
- *FUNDING: OK
- *AUTHOR CONTRIBUTIONS: OK
- *DisclCIS: Please rename to "Disclosure and Competing Interests Statement"
- *REFERENCE FORMAT: Please correct to EMBO style: In the text of the manuscript, a reference should be cited by author and year of publication; no more than two authors may be cited per reference; 'et al' should be used if there are more than two authors (i.e. Smith & Jones, 2003; Smith et al, 2000). In the reference list, citations should be listed in alphabetical order and then chronologically, with the authors' surnames and initials inverted; where there are more than 10 authors on a paper, 10 will be listed, followed by 'et al.'. Please remove the dois for published articles.
- *DATASET EV LEGENDS:
 - Please rename the datasets "Dataset EV1" - EV6 and upload each dataset as a separate file
 - Please remove the suppl. tables from the manuscript text, rename them "Tabl EV1" - EV4 and upload each table as a separate file.
- *APPENDIX 1 FILE WITH ToC: n/a OR see below
- *REAGENT TABLE: missing. Please download the template from the author guidelines, complete the form and upload it as a separate file.
- *DATA NOT SHOWN: OK
- *SYNOPSIS IMAGE: Please provide a visual abstract as a high-resolution jpeg file 550 px-wide x 300-600 pixels high to illustrate your article.
- *SYNOPSIS TEXT: Please provide a short standfirst (maximum of 300 characters, including space) as well as 2-5 one sentence bullet points that summarise the paper as a .doc file. Please write the bullet points to summarise the key NEW findings. They should be designed to be complementary to the abstract - i.e. not repeat the same text. We encourage inclusion of key acronyms and quantitative information (maximum of 30 words / bullet point).
- *FIGURE CALLOUTS: Please ensure that each figure panel and dataset is cited and in sequential order

Additional Notes:

- Please move the Methods after the Discussion section
- Please rename the "Supplementary Alignment Files" to "Dataset EV7" and provide the files in .aln, .phy or .fa format instead of PDF, then upload them as one zip file. Please add a citation in the manuscript text.

DATA CHECK:

- Data Availability section:
Please note that the data availability statement is not provided in the manuscript. Please provide.
- Figure legends:
 1. Please define the annotated p values ****/****/**/* as well as provide the exact p-values for the same in the legend of figure 3A as appropriate.
 2. Please note that the exact p values are not provided in the legends of figures 3B, 5; S4. Please provide.
 3. Please indicate the statistical test used for data analysis in the legend of figure 3A.
 4. Please note that information related to n is missing in the legends of figures 3C, S4. Please provide.
 5. Please note that the error bars are not defined in the legend of figure 3C. Please define in figure legend.

Specific comments by Referees

Referee #1:

The authors have well addresses questions raised, and I have no further comments

Referee #2:

Having reviewed this previously through Creative Commons, I am happy with the responses provided by the authors to my previous review

Rev_Com_number: RC-2025-03068

New_manu_number: EMBOJ-2025-122545-T

Corr_author: Champer

Title: Assessing target genes for homing suppression gene drive

The authors addressed the remaining editorial issues.

Dear Dr. Champer,

I am pleased to inform you that your manuscript has been accepted for publication in the EMBO Journal.

Yours sincerely,

Yehu Moran
Academic Editor
The EMBO Journal

Please note that it is The EMBO Journal policy for the transcript of the editorial process (containing referee reports and your response letters) to be published as an online supplement to each paper. If you should prefer removal of any referee-only figures included in the point-by-point response(s), e.g. because they may still be used for future publication or because they have been reproduced from published work by others, please do let us know immediately via response email.

More information is available here: https://www.embopress.org/transparent-process#Review_Process